

# FROSTBYTE: A reproducible data-driven workflow for probabilistic seasonal streamflow forecasting in snow-fed river basins across North America

Louise Arnal[1,a], Martyn P. Clark[1,2], Alain Pietroniro[2], Vincent Vionnet[3], David R. Casson[2], Paul H. Whitfield[1], Vincent Fortin[3], Andrew W. Wood[4,5], Wouter J. M. Knoben[2], Brandi W. Newton[6], and Colleen Walford[7]

[1]University of Saskatchewan, Centre for Hydrology, Coldwater Laboratory, Canmore, AB, Canada
[2]Department of Civil Engineering, University of Calgary, Calgary, AB, Canada
[3]Meteorological Research Division, Environment and Climate Change Canada, Dorval, QC, Canada
[4]National Center for Atmospheric Research, Boulder, CO, USA
[5]Colorado School of Mines, Golden, CO, USA
[6]Airshed and Watershed Stewardship Branch, Alberta Environment and Protected Areas, Calgary, AB, Canada
[7]Alberta River Forecast Center, Environment and Protected Areas, Government of Alberta, Edmonton, AB, Canada
[a]now at: Ouranos, Montréal, Québec, Canada

**Correspondence:** Louise Arnal (louise.arnal@usask.ca)

**Abstract.** Seasonal streamflow forecasts provide key information for decision-making in sectors such as water supply management, hydropower generation, and irrigation scheduling. The predictability of streamflow on seasonal timescales relies heavily on initial hydrological conditions, such as the presence of snow and the availability of soil moisture. In high-latitude and high-altitude headwater basins in North America, snowmelt serves as the primary source of runoff generation. This study

presents and evaluates a data-driven workflow for probabilistic seasonal streamflow forecasting in snow-fed river basins across North America (Canada and the USA). The workflow employs snow water equivalent (SWE) measurements as predictors and streamflow observations as predictands. Gap filling of SWE datasets is accomplished using quantile mapping from neighboring SWE and precipitation stations, and Principal Component Analysis is used to identify independent predictor components. These components are then utilized in a regression model to generate ensemble hindcasts of streamflow volumes for 75 nival

basins with limited regulation from 1979 to 2021, encompassing diverse geographies and climates. Using a hindcast evaluation approach that is user-oriented provides key insights for snow monitoring experts, forecasters, decision-makers, and workflow developers. The analysis presented here unveils a wide spectrum of predictability and offers a glimpse into potential future changes in predictability. Late-season snowpack emerges as a key factor for predicting spring/summer volumes, while high precipitation during the target period presents challenges to forecast skill and streamflow predictability. Notably, we can pre-

dict lower and higher than normal streamflows during the spring to early summer with up to five months lead time in some basins. Our workflow is available on GitHub as a collection of Jupyter Notebooks, facilitating broader applications in cold regions and contributing to the ongoing advancement of methodologies.



# 1 Introduction

Seasonal streamflow forecasts play an important role in various sectors, including water supply management, hydropower
generation, and irrigation scheduling. It can also provide early warning of floods and droughts. Around the globe, a diverse
range of predictors plays a crucial role in seasonal streamflow forecasting. This includes antecedent hydrological conditions
(e.g., snowpack, past streamflow, soil moisture) and future conditions (e.g., future precipitation, climate signals). See Yuan
et al. (2015) for a comprehensive review of the dominant sources of seasonal hydrological predictability. Various forecasting
methods leverage these predictors and hydrological processes that drive streamflow variability in regions of interest.

In Canada and much of the USA, snowmelt is an important driver of streamflow. In spring, the snow accumulated during
winter serves as a substantial water reservoir in high-altitude mountainous regions, often referred to as "water towers" (Viviroli
et al., 2007). Gradually, this natural water storage releases its stored contents downstream to the rivers through the process of
snowmelt. In the western USA, operational seasonal hydrological forecasting relies on the long-term predictability provided
by winter snow conditions (Wood et al., 2016). This important natural water supply is however threatened by climate change.
Immerzeel et al. (2020) assessed the vulnerability of the world's water towers and found that in North America, vulnerabilities
are associated with both population growth and rising temperatures. By understanding the predictability of streamflow origi-
nating from snowmelt, we can better address the challenges posed by climate change and effectively manage these invaluable
water sources for the future.

Over the past few decades, significant advances have been made in our understanding of forecast quality and hydro-
meteorological predictability on seasonal timescales. These have been facilitated, in part, by the continuous improvements in
technological capabilities. As a result, a wide range of approaches now exists for streamflow forecasting on seasonal timescales,
including process-based, data-driven, and hybrid models, each possessing distinct advantages and limitations (Slater et al.,
2023). This paper focuses on data-driven approaches.

Data-driven forecasting involves predicting a variable of interest (known as predictand; e.g., streamflow spring volume) by
establishing relationships between the predictand and one or more predictors (e.g., snowpack, past streamflow, climate sig-
nals). Various techniques can be employed to model these relationships, ranging from simple linear regressions to more com-
plex machine learning (ML)/artificial intelligence (AI) methods. Consider the following noteworthy data-driven approaches
for seasonal streamflow forecasting: i) Principal Component Regressions (PCR) have proven effective in streamflow volume
forecasting in the USA (Garen, David C., 1992; Mendoza et al., 2017; Fleming and Garen, 2022); ii) Bayesian joint proba-
bility statistical modelling has demonstrated its capability for ensemble seasonal streamflow forecasting in Australia (Wang
et al., 2009); iii) Seven different Generalized Additive Models for Location, Scale and Shape statistical models were tested to
forecast quantiles of seasonal streamflow in the Midwest USA, using a range of predictors such as precipitation, temperature,
agricultural land cover and population (Slater and Villarini, 2017); iv) A robust M-regression model was first tested for hydro-
logical forecasting for ensemble seasonal streamflow forecasting in the South Saskatchewan River Basin (Canada), extending
the operational forecast lead time by up to two months (Gobena and Gan, 2009); v) Regression models were applied for winter
and early spring streamflow forecasting in large North American river basins in Canada and the USA, based on snowpack





information (Dyer, 2008); vi) Machine learning (ML)/artificial intelligence (AI) is now increasingly explored for this type of application. Fleming et al. (2021) explored the use of AI for forecasting of water supplies in the western USA. They showed that it meets the quality and technical feasibility requirements for operational adoption at the US Department of Agriculture

Natural Resources Conservation Service (NRCS).

This work builds on the literature and addresses research gaps by extending the spatial domain of previous studies to include both Canada and the USA. In this work, we use PCR to predict future streamflow from snow water equivalent (SWE) information as the sole predictor given its importance for seasonal streamflow prediction. PCR stands as a well-established and widely used data-driven method, as demonstrated by the non-exhaustive list above. Simple statistical regression methods such as PCR

offer several key advantages, including their local applicability, intuitive nature (i.e., use of local data to represent known and observed hydrological processes locally), speed and low computational resource requirements. These methods are additionally straightforward to implement and potentially highly reliable.

Mendoza et al. (2017) showed that increasing methodological complexity (in their study this was defined as the gradient from purely data-driven techniques to the use of process-based models) does not always lead to improved forecasts. Empha-

sizing simplicity in modeling provides a robust foundation for enhancing our comprehension of hydrological processes and supports ongoing improvements to forecast quality (including through model developments and the use of new observations), as highlighted by Delgado-Ramos and Hervas-Gamez (2018). This approach additionally supports reproducibility to enable collaborative advancements through open science practices (Knoben et al., 2022).

In this paper, we present a reproducible data-driven workflow designed for probabilistic streamflow forecasting in nival

(i.e., snowmelt-driven) river basins across Canada and the USA (Section 2.2). For the sake of simplicity, we use the term 'North America' to refer to the forecasting domain used in this study. Through the analysis of the hindcasts produced with this workflow, we address the research question: can SWE be used as a reliable predictor of future streamflow in nival river basins across North America (Section 3)? In the discussion of our findings, we extract essential insights relevant to snow monitoring experts, forecasters, decision-makers, and workflow developers, addressing an important research gap in knowledge translation

(Section 4).

## 2  Data and methods

### 2.1  Data

Four types of data are needed to run the workflow for North American river basins. These include river basin shapefiles and station data for streamflow, SWE and precipitation (Fig. 1). Each data type is explained in the following sections.

### 2.1.1  Basin shapefiles and streamflow data

For the USA, we use shapefiles and streamflow observations for basins with limited regulation from the USGS Hydro-Climatic Data Network 2009 (HCDN-2009; Lins, 2012; Falcone, 2011). For Canada, we use shapefiles and streamflow observations







**Figure 1.** Maps of basin outlines (a) and input station data - i.e., streamflow (b), SWE (c), and precipitation (d) - across North America. Note that there are fewer streamflow stations than basin outlines as map (b) only shows streamflow stations with data for the period 1979–2021. Additionally, map (d) shows all SCDNA stations available. This includes stations with temperature data but no precipitation data, that were not used in this study.

for basins with limited regulation from the Water Survey of Canada (WSC) HYDAT Reference Hydrometric Basin Network (RHBN) subset, called RHBN-N (ECCC, 2021). The reference hydrologic networks include only stations considered to have





good quality data and were screened to avoid the influences of regulation, diversions, or land use change (Whitfield et al.,
2012). RHBN-N was created to provide a nationally balanced network suitable for national studies.

We downloaded streamflow data for the period 1979-01-01 to 2021-12-31 as data for this period were available for many stations in the dataset, and this was deemed an appropriate time series length for the purpose of this study. Data for the USA were downloaded from the National Water Information System (NWIS; USGS) using the Python package dataretrieval (Hodson and Hariharan, 2023). Data for Canada were downloaded from the WSC HYDAT database (ECCC, 2018) using the EASYMORE Python package (Gharari et al., 2023). See Fig. 1 (a) and (b) for maps of the basin shapefiles and streamflow stations that were retained.

### 2.1.2 Snow Water Equivalent data

Snow Water Equivalent (SWE) measurements were downloaded for the period 1979-10-01 to 2022-07-31. For Canada, measurements are from the Canadian historical Snow Water Equivalent dataset (CanSWE, Version V5; Vionnet et al., 2021b), available on Zenodo (Vionnet et al., 2023). CanSWE is a database of SWE data collected from numerous provincial/territorial, academic, and other agencies across Canada. Other measurements are from the Ministère de l'Environnement, de la Lutte contre les changements climatiques, de la Faune et des Parcs (MELCCFP) in Québec (Canada) and cannot be shared publicly. For the USA, measurements are mainly from the Natural Resources Conservation Service (NRCS) manual snow surveys and SNOTEL automatic snow pillows. The NRCS snow courses can be downloaded using the following GitHub repository: https://github.com/CH-Earth/snowcourse. For the SNOTEL, we use data from the bias-corrected and quality-controlled (BCQC) dataset from the Pacific Northwest National Laboratory (PNNL; https://www.pnnl.gov/data-products; Sun et al., 2019; Yan et al., 2018). Figure 1 (c) shows a map of the SWE stations used for this analysis. Note that snow data are missing in the northern central part of the USA, and a viable substitute in the future could be utilizing airborne gamma snow data (Cho et al., 2020).

We use all available streamflow and SWE data and do not filter out data values based on their quality flags. The reason is that we perform gap filling of all time series within the workflow, and we trust that data providers are the most competent individuals to handle the initial infilling. We invite readers to refer to these datasets for a list of quality flags.

### 2.1.3 Precipitation data

Precipitation station data were downloaded from the Serially Complete Dataset for North America (SCDNA, Version 1.1) for the period 1979-01-01 to 2018-12-31 (Tang et al., 2020a). The SCDNA dataset is available on Zenodo (Tang et al., 2020b). Figure 1 (d) shows a map of the SCDNA stations.

### 2.2 Methods: Workflow

The workflow developed is structured in five Jupyter Notebooks: 1) Regime classification, 2) Streamflow pre-processing, 3) SWE pre-processing, 4) Forecasting, and 5) Hindcast verification. Each Notebook is coded in Python, and provides concise



descriptions of its purpose, decisions, and underlying assumptions, whenever necessary, as well as a step-by-step overview of the annotated code, accompanied by visuals. The workflow, called FROSTBYTE (Forecasting River Outlooks from Snow Timeseries: Building Yearly Targeted Ensembles), is available on GitHub (Arnal et al., 2023a, v0.9.0). Note that the data downloading step (see Section 2.1) was not included in the GitHub workflow. Instead, sample data are provided for the Bow River at Banff (Alberta, Canada) and for the Crystal River Abv Avalanche Crk, Near Redstone (Colorado, USA). Figure 2 provides a visual of the methods for each Jupyter Notebook. These will be described in more details in the sections below.





**Figure 2.** Detailed graphical methods for each Jupyter Notebook of the FROSTBYTE workflow.



### 2.2.1 Regime classification: Basins selection

To ensure the feasibility of producing forecasts using PCR from SWE predictors, we impose the following requirements on the basins used in this study:

- The basin must have a nival regime. Discussed in more detail in this section.

- The basin must contain at least one SWE station.

- The basin must have at least 20 years of overlapping SWE and streamflow data (regardless of whether the years are complete as the data are further pre-processed to fill gaps, see Sections 2.2.2 and 2.2.3). If the basin contains multiple SWE stations, only one station needs to fulfill this requirement.

The river basins in the USA and Canada for which we collected data in the previous step (see Section 2.1) are subject to a wide range of hydroclimatic conditions. In this step, we further subset these basins to only keep basins that experience nival regimes - i.e., basins for which we can reasonably expect SWE to hold substantial predictability for streamflow forecasting. The existence of nival regimes can be inferred from climate classification schemes that account for the fraction of precipitation falling as snow in a given place. However, we instead opt to use an approach that identifies the typical flow regime for each
basin based directly on the observed streamflow in that basin.

To classify the streamflow regimes, we used circular statistics (Burn et al., 2010). Circular statistics measure the timing and regularity of hydrological events such as flow peaks. For this study, three different streamflow peak metrics were used to provide more robust results than using a single metric, because strengths of one metric can compensate for weaknesses in another. Some of these weaknesses are discussed in Court (1962), Whitfield (2013), and Burn and Whitfield (2017).

The metrics used to identify peak flow events are [a] the streamflow annual maxima, [b] the streamflow peaks over threshold, and [c] the streamflow centre of mass (i.e., date on which 50% of the water-year streamflow occurs; Court, 1962). For the peak over threshold metric [b], the threshold was defined as the smallest annual maximum streamflow observed during the historical period in each basin. All metrics were computed using a water year from October $1^{st}$ to September $30^{th}$ of the following calendar year to link winter snow accumulation to the current year snowmelt. In order to maximize the amount of available
data, we first performed gap filling through linear interpolation of the daily streamflow data. More information about the interpolation can be found in Section 2.2.2. Note that the streamflow interpolation was performed twice independently, once prior to the regime classification and once as part of the streamflow pre-processing, for a more logical flow of the workflow. The streamflow annual maxima [a] and the center of mass [c] metrics required complete years of data (i.e., any water year with missing observations was discarded), while the peak over threshold metric [b] allowed for incomplete years of data to identify
peak flow events.

For each metric, we then calculated the average date of occurrence for these peak flow events by determining the circular mean of all event dates. Additionally, we assessed the regularity of the peaks by calculating the spread in the dates of occurrence of these events. The regularity value, which ranges between zero and one, provides a measure of how consistent the event dates are. Larger values indicate a higher level of regularity in the dates. Equations used for the regime classification can be found



in the Appendix. For each metric, we identified nival basins as those with an average date of occurrence of peak flow events between March $1^{st}$ and August $1^{st}$ and a regularity above or equal to 0.65 (defined based on results presented in Burn and Whitfield, 2023). We finally selected all nival basins identified by the three individual metrics.

### 2.2.2    Streamflow pre-processing

We processed the daily streamflow data for all previously identified nival basins (see Section 2.2.1) and converted them into
volumes that capture the spring freshet and that may be of interest of water users (e.g., for water supply management, hydropower generation, irrigation scheduling, early warnings of floods and droughts). These volumes serve as the predictands for the forecasting process (see Section 2.2.4).

We first performed gap filling through linear interpolation of the daily streamflow data in order to maximize the amount of available data. The maximum allowable gap length for interpolation was set to 15 days, consistent with the SWE interpolation
approach (see Section 2.2.3).

We then computed volumes for periods without any missing data for each nival basin: January $1^{st}$ to September $30^{th}$, February $1^{st}$ to September $30^{th}$, March $1^{st}$ to September $30^{th}$, etc., until September $1^{st}$ to $30^{th}$. Volumes are calculated by summing the daily streamflow observations over the time periods mentioned above. These volume aggregation periods will be referred to as 'target periods' in the context of forecasting throughout this paper. The volumes dataset was saved for all basins
as a NetCDF file.

### 2.2.3    SWE pre-processing

For each previously identified nival basin (see Section 2.2.1), we processed the SWE data to fill gaps as the subsequent Principal Component Analysis (PCA) does not allow for missing values. These pre-processed SWE data serve as the predictors for the forecasting process (see Section 2.2.4).
We selected SWE and precipitation (if any) stations located in each nival basin. The precipitation data is used to maximize the amount of data available as predictors. It was accumulated over water years for each precipitation station to serve as a proxy for SWE. To further enhance the availability of data, we applied linear interpolation to fill gaps in the daily SWE records. The maximum allowable gap length for interpolation was set to 15 days, which aligns with the streamflow interpolation approach described in Section 2.2.2 and with the window of +/- 7 days used in the subsequent steps.
Subsequently, statistics necessary for the next gap filling steps were calculated for all extracted SWE and precipitation stations. Namely, a cumulative distribution function (CDF) was constructed for station for each day of the year (+/- 7 days). A CDF could only be constructed when at least ten data points were available. Spearman's rank correlation coefficients were calculated between each basin's SWE and precipitation station for each day of the year (+/- 7 days). Correlations could only be calculated when a minimum of three data points were available. It is important to note that the minimum sample size criteria
for the CDF and the correlation calculations are user-defined. For this study, they were set to the values mentioned above in order to balance the need for a sufficiently large sample size for reliable results with the goal of filling in as many gaps as possible. The impact of these decisions could be explored in future research.



We then perform gap filling using quantile mapping. Looping over SWE stations, for each missing SWE data point in a target station (i.e., the station requiring gap filling), a suitable SWE/precipitation donor station (i.e., the station providing data for infilling the target station's gap) was identified based on the following criteria:


– The donor station must have data on or around the date to be filled (within a window of +/- 7 days).

– The target and the donor stations should have a constructed CDF for the day of the year corresponding to the date to fill.

– The correlation between the target station and the donor station should be the highest amongst all potential donor stations and exceed a minimum correlation threshold of 0.6. Stations with correlations larger than but close to 0.5 could be deemed as only marginally correlated. We require a strong positive correlation to ensure the quality of the gap filling process and set the threshold to 0.6 for a station to be accepted as a donor station.


Based on these criteria, the value from the donor station closest to the date requiring filling is used to estimate the target station's value on the missing value's date. Note that the automatic SWE stations have a higher temporal frequency than the manual snow surveys, which could make the automatic stations preferable as potential donor stations.

As a result, a new gap-filled SWE dataset was generated and saved for each nival basin as a NetCDF file. Estimated values were clearly distinguished from the original observations using a specific flag.

Additionally, we developed an artificial gap filling function to enable users to assess the quality of the gap filling process. Results are shown for the Bow River at Banff (Alberta, Canada), one of the workflow testbeds, in the Appendix (Fig. A1 and A2). We do not show results for all other river basins as the artificial gap filling is not the primary focus of this study.

### 2.2.4  Forecasting

Using the pre-processed predictors and predictands (see Sections 2.2.2 and 2.2.3) as inputs to an Ordinary Least Squares (OLS) regression model, we generate time series of ensemble hindcasts for each nival basin. Hindcasts are initialized on the first day of each month between January $1^{st}$ and September $1^{st}$ (both ends included) for target periods January $1^{st}$ to September $30^{th}$, February $1^{st}$ to September $30^{th}$, March $1^{st}$ to September $30^{th}$, etc., until September $1^{st}$ to $30^{th}$.

For each initialization date-target period combination, we first remove all years that have any missing values in the predictand and/or predictor datasets. We require a minimum of eleven years of overlapping data, comprising ten years for training the regression model and an additional year for generating the hindcast, using the leave-one-out cross-validation approach. Consequently, we may not be able to generate hindcasts for specific nival basins previously identified in Section 2.2.1, and for specific initialization date-target period combinations.

We then transform the gap-filled SWE from Section 2.2.3 into principal components (PC) to eliminate any intercorrelation amongst the SWE stations (Garen, David C., 1992). Principal Component Analysis (PCA) is a statistical method used to transform a set of intercorrelated variables into an equal number of uncorrelated variables. This step becomes particularly essential after gap filling, which might have introduced additional correlation among the SWE stations. In addition, the PCA is central to characterizing the spatio-temporal variability of the predictor variable. The first PC (i.e., the PC which captures





most of the total variance in the set of variables) serves as the predictor for the forecasting process. We only select the first principal component in order to avoid any overfitting that could be caused by having too many predictors and a short time series. In a subseasonal climate forecasting study, Baker et al. (2020) showed that even using only the first two PCs could lead to overfitting for many river basins of the contiguous USA. We acknowledge however that this is a topic that warrants further exploration and discuss this in more detail in Section 4.4. We conduct a PCA and fit a new model for each predictor-predictand

combination.

We subsequently split the predictor-predictand data using a leave-one-out cross-validation approach. We fit an Ordinary Least Squares (OLS) regression model on all years available for training. We then apply this model to the predictor year that was excluded, resulting in a deterministic volume hindcast for the corresponding target period. An ensemble of 100 members is then generated from this deterministic hindcast by drawing random samples from a normal (Gaussian) distribution. The distribution

has a mean of zero and a standard deviation equal to the root mean squared error (RMSE) between the deterministic hindcasts and observations during the training period. We repeat this step until ensemble hindcasts have been generated for all years in the predictor-predictand dataset. It is important to note that hindcasts were generated only if there were at least ten years of overlapping predictand-predictor training data for a given initialization date-target period combination.

Once we have generated hindcast time series for all initialization date-target period combination and all nival basins (when

possible), each basin's hindcasts are individually saved in a new NetCDF file.

### 2.2.5   Hindcast verification

An overview of the various deterministic and probabilistic metrics used to assess the quality of the hindcasts, and what they measure, is provided in Table 1. To quantify sampling uncertainty, all metrics are computed using bootstrapping (Clark et al., 2021b). We draw 100 random samples of hindcast-observation pairs, with replacement.

To enable a meaningful comparison of performance across different basins, we defined specific target periods that capture each basin's peak flow (Qmax). We refer to these periods as 'periods of interest' throughout the paper. For each nival basin, the period of interest begins in the month of the basin's Qmax and extends until the end of the water year. For example, if a basin has its Qmax on May $15^{th}$, its period of interest will be May $1^{st}$ to September $30^{th}$.

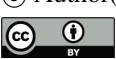



**Table 1.** Overview of the various deterministic and probabilistic metrics used to assess the quality of the hindcasts.

| Metric | Description | Equation |
|---|---|---|
| **KGE''** | The Kling-Gupta Efficiency (KGE) is a deterministic and combined measure of the forecast correlation, bias and variability (Gupta et al., 2009). The "KGE'' was proposed by Tang et al. (2021) to solve issues arising with 0 values in the KGE or KGE'. | $KGE'' = 1 - \sqrt{\beta^2 + (\alpha - 1)^2 + (\rho - 1)^2}$ <br><br> KGE'' is unitless and ranges between $-\infty$ and 1, with a perfect score of 1. This metric was computed on the ensemble hindcast medians. Equations for the components of the KGE'' can be found in the Appendix. |
| **RI** | The reliability index (RI) is a probabilistic measure of the forecast reliability - i.e., the adequacy of the forecast ensemble spread to represent the uncertainty in the observations. More specifically, it measures the average agreement between the predictive distribution and the observations by quantifying the closeness between the empirical CDF of the ensemble forecast with the CDF of a uniform distribution (Renard et al., 2010; Mendoza et al., 2017). | $RI = 1 - 2\alpha' = 1 - 2\left[\frac{1}{N}\sum_{i=1}^{N}|P_i(o_i) - U_i(o_i)|\right]$ <br><br> RI is unitless and ranges between 0 and 1, with a perfect score of 1. Note that $\alpha$ is often used as a symbol for the reliability index. Here, we use the notation RI instead so as not to confuse this metric with the KGE'' variability ratio ($\alpha$). <br><br>  |
| **CRPSS** | The Continuous Rank Probability Skill Score (CRPSS) is a probabilistic measure of the forecast skill - i.e., the performance (in terms of the CRPS) of the ensemble forecast against a baseline (here, the observed climatology). The CRPS is a measure of the difference between the predicted and the observed cumulative distribution functions (CDF) (Hersbach, 2000). | $CRPSS = 1 - \frac{CRPS_{forecast}}{CRPS_{baseline}}$ <br><br> where $CRPS = \frac{1}{N}\sum_{i=1}^{N}\int_{-\infty}^{\infty}[F(q) - Fo(q)]^2 dq$ <br><br> CRPSS is unitless and ranges between $-\infty$ and 1, with a perfect score of 1. CRPSS=0 represents the threshold below which hindcasts have no skill compared to the baseline. |
| **ROC** | The Relative Operating Characteristic (ROC) is a probabilistic measure of the forecast resolution - i.e., the ability of the forecast to discriminate between events and non-events (Mason and Graham, 2002). Here, the ROC was computed for low/high flows (i.e., flows below/above the lower/upper tercile of the observed climatology). This metric can serve as an indicator of the forecast's 'potential usefulness' and is of particular importance for decision-makers (Arnal et al., 2018; Emerton et al., 2018). | $POD = \frac{hits}{hits + misses}$ <br><br> $POFD = \frac{false\ alarms}{correct\ negatives + false\ alarms}$ <br><br> The ROC area under the curve (or ROC AUC) is unitless and ranges between 0 and 1, with a perfect score of 1. ROC AUC=0.5 represents the threshold below which hindcasts have no skill. |

F(q): ensemble forecast CDF. Fo(q): observations CDF. $P_i(o_i)$: empirical CDF of the ensemble forecast p-values for year i. $U_i(o_i)$: uniform distribution U[0,1].



To guide the hindcasts' evaluation, we formulate two hypotheses regarding the hindcasts generated using this approach:

1. The hindcast performance is expected to be better for hindcasts initialized around the peak SWE in each basin.

    2. Higher hindcast quality is anticipated for hindcasts with high antecedent SWE content and low precipitation during the hindcast target period in each basin.

To support the interpretation of the hindcast evaluation and verify those hypotheses, we computed two additional measures. To evaluate the first hypothesis, we calculated the 'SWE content' for all initialization dates (i.e., note that forecasts were
initialized on the $1^{st}$ of each month between January and September for computational reasons, and that we may as a result miss peak SWE). We calculated the median percentage of maximum SWE for each initialization date across all available years of data for all SWE stations. The maximum SWE value was determined for each water year for this calculation. E.g., A median percentage of maximum SWE of 50% indicates that across all years for which we have data for a given SWE station, in the median year, half of that year's maximum SWE is present at that location on the forecast initialization date. The equation to
calculate the 'SWE content' can be found in the Appendix.

To evaluate the second hypothesis, we computed the ratio of precipitation to SWE (i.e., P/SWE). To achieve this, we followed these steps for each basin:

    – We calculated the precipitation accumulation for each year, target period, and each precipitation station within the basin. We then calculated the climatological medians for precipitation accumulation, considering each station and target pe-
riod, and subsequently averaged them over the entire basin. This gave us the basin-averaged precipitation accumulation climatological medians for each target period.

    – We calculated the SWE climatological median for each initialization date and each SWE station within the basin. We then averaged the SWE climatological medians over the entire basin, resulting in basin-averaged SWE climatological medians for each initialization date.

– Finally, we divided the basin-averaged precipitation statistics by the corresponding basin-averaged SWE statistics for each combination of initialization date and target period.

## 3   Hindcast evaluation

In this section, we quantify the range of predictability for 62 of the 75 identified nival basins across North America (Fig. 3) by analyzing results for various deterministic and probabilistic metrics, as outlined in Section 2.2.5. A total of 13 basins
were excluded from this analysis as no hindcasts could be generated for those basins given a limited amount of overlapping predictor-predictand training data (as outlined in Section 2.2.4).

The figures presented in the sub-sections below display only the bootstrapping means. The corresponding bootstrapping ranges, showing the uncertainty in these estimates, can be found in the Appendix (Fig. A4).





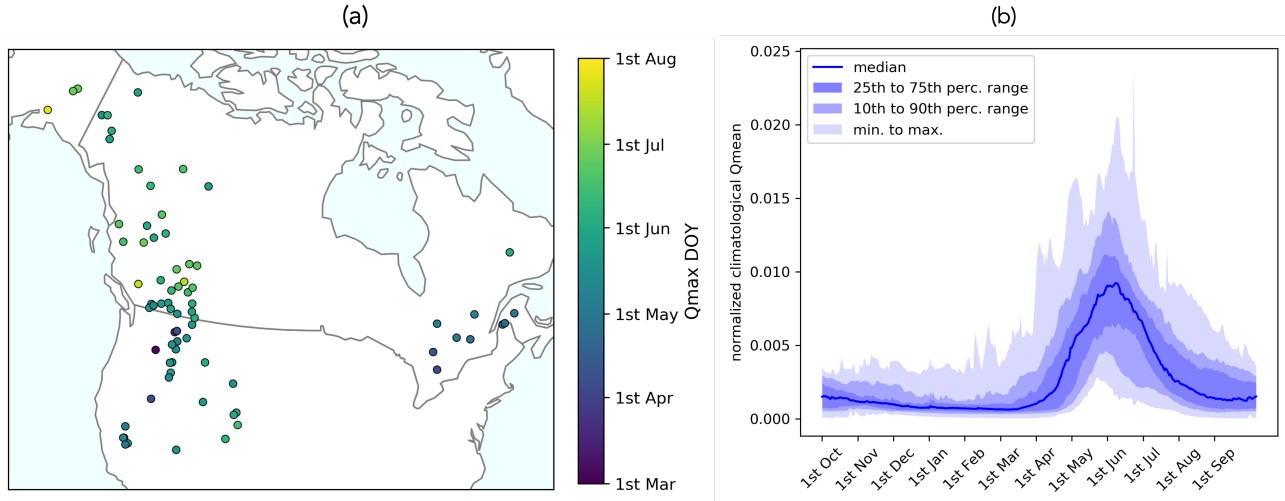

**Figure 3.** Map and hydrograph of the 75 nival basins with limited regulation that meet the data requirements. Basins identified as having a nival regime that did not meet the data requirements are shown in the Appendix (Fig. A3). (a) The map shows the average day of the year (DOY) when the maximum streamflow (Qmax) occurs for each nival basin. (b) The hydrograph displays the normalized climatological mean streamflow for all 75 nival basins, with the median across all basins as the blue line and the variation in responses across all basins indicated by the shaded percentiles. The climatological mean streamflow timeseries for each basin was normalized by dividing it by the sum of the timeseries.

## 3.1 Correlation, bias, and variability

Fig. 4 shows the hindcast performance in terms of the Kling-Gupta Efficiency (KGE") and its decomposition into correlation, variability, and bias, as a function of hindcast initialization dates. The KGE" can vary significantly across different target periods, and these differences tend to increase with later initialization dates. This highlights the impact of both target periods and model initialization on the hindcast quality. Overall, the KGE" is higher for early target periods and decreases with later target periods. This hints that the snowpack holds less predictability as we move from the spring to the summer/fall months, 280   and may be an indication of a shift from snow to rain as the dominant driver of streamflow.

For hindcasts generated for target periods January $1^{st}$ to September $30^{th}$ until June $1^{st}$ to September $30^{th}$, the KGE" increases towards the perfect value as we are approaching the start of the target period being predicted (i.e., with 0 lead months - e.g., hindcasts for June $1^{st}$ to September $30^{th}$ initialized on June $1^{st}$). Later target periods (August $1^{st}$ to September $30^{th}$ and September $1^{st}$ to $30^{th}$) show a declining KGE" overall. Hindcasts for July $1^{st}$ to September $30^{th}$ show a mixed signal: they 285   follow the later target periods' curves but peak for the June $1^{st}$ initialization, after which they quickly decline.

The correlation and the variability ratio show similar patterns to the KGE". The variability tends to be underestimated overall. This may be a direct consequence of using only PC1 as a predictor, although further comprehensive testing would be required





to confirm this. The bias ratio is overall slightly positive, indicating that the hindcast medians overall overestimate the observed volumes. This is especially noticeable for hindcasts initialized between July $1^{st}$ and September $1^{st}$.

**Figure 4.** KGE" of the hindcast medians and its decomposition (i.e., correlation, variability, and bias) as a function of hindcast initialization dates. Each line displays median values across all basins for each target period. On all plots, the dashed lines represent the perfect value for each metric.

## 3.2 Reliability

Fig. 5 shows the hindcast reliability as a function of hindcast initialization dates. From the literature, we expect the hindcasts generated to have high reliability given the ensemble generation approach used (i.e., statistical analysis of errors in cross-validated hindcasts, compared to other methods, such as using an ensemble of models or an ensemble meteorological inputs without any pre-processing). Indeed, overall, hindcasts display high reliability across all river basins, initialization dates, and target periods.



The reliability of hindcasts is not entirely perfect, primarily due to the leave-one-out cross-validation approach used. We expect this effect to be more noticeable when the sample size is smaller, and hypothesize that it may partially account for the decrease in hindcast reliability with increasing initialization dates observed here.

The lower reliability for the September $1^{st}$ to $30^{th}$ target period additionally provides further support for the diminishing
significance of snow information during periods characterized by non-snowmelt-driven flow.

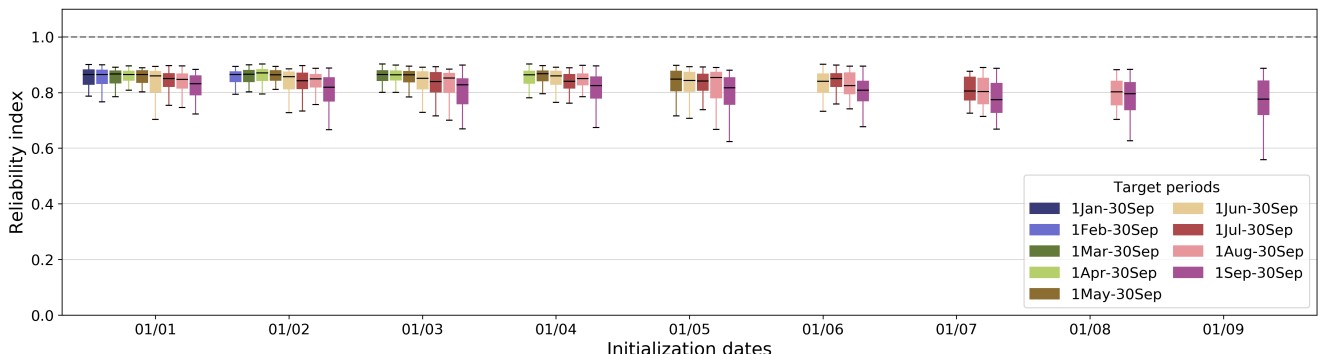

**Figure 5.** Hindcast reliability as a function of hindcast initialization dates. The boxplots display values for all basins. The dashed line represents the perfect value.

## 3.3 Skill

Fig. 6 shows the hindcast skill in terms of the Continuous Rank Probability Skill Score (CRPSS), as a function of hindcast initialization dates. On average, hindcasts are as good as the baseline when they are initialized on January $1^{st}$. They gradually get more skilful (i.e., better than the observed streamflow climatology) for initialization dates between February $1^{st}$ and June $1^{st}$.
Beyond June $1^{st}$, hindcasts for the summer/fall target periods exhibit no overall skill (i.e., worse than the observed streamflow climatology). Overall, earlier target periods have better skill than the later target periods. This is similar to the pattern observed for the KGE" and again hints at a shift from snow to rain as the dominant driver of streamflow. It further suggests that as we approach peak SWE (see Fig. 9 (a) in Section 4.1), we can extract more valuable information and enhance the hindcast skill.

This SWE-based forecasting approach is unskilful with later initialization dates and target periods, meaning that using the
streamflow climatology provides better results than using this approach. Note that the CRPSS results might be impacted by the ensemble size of the hindcasts (100 members) compared to the ensemble size of the baseline (the number of members equals the number of years in the climatology, excluding the year being forecasted, and varies across basins and target periods).

As shown by the boxplots' span, the CRPSS can vary considerably across different basins. This implies that the predictive performance might differ significantly depending on the geographical location. To explore the geographical distribution of the
CRPSS, we show maps of the CRPSS with zero to six months lead time (Fig. 7). Overall, the CRPSS decreases with increasing lead time, and by three to four month lead time the skill is on average marginal. Note that some basins may show increasing





skill with increasing lead time, or a more complex picture, highlighting the intricate interplay between initialization date and target period.

Results are very variable across space, and some river basins already show low to no or negative skill throughout all lead months, and the skill drops quickly after zero months lead time. These are mostly basins situated in the northwest and in the east. Pockets of higher skill are seen across several lead months for basins situated in western North America, in and and around the Rocky Mountains and the Sierra Nevada mountain ranges. On the other hand, some basins display low to no skill throughout all initialization dates. We speculate that basins exhibiting higher skill are those characterized by substantial contributions of SWE to streamflow predictability and substantial year-to-year variability, thereby enhancing skill in comparison to the climatological reference.

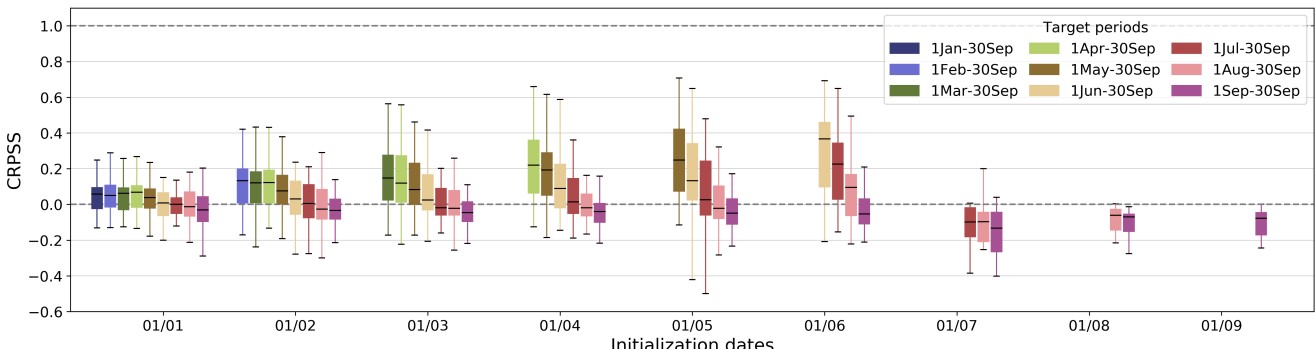

**Figure 6.** Hindcast CRPSS for each target period as a function of hindcast initialization dates. The boxplots display values for all basins. The upper dashed line (CRPSS=1) represents the perfect value and the lower dashed line (CRPSS=0) represents the threshold below which hindcasts have no skill compared to streamflow climatology.

## 3.4 Potential usefulness

Fig. 8 shows the hindcast potential usefulness in terms of the Relative Operating Characteristic area under the curve (ROC AUC), as a function of hindcast initialization dates, for predicting (a) high flows and (b) low flows. Unlike plots for the KGE" and its decomposition and for the CRPSS, these plots show results for each basin's period of interest only, as we are interested in understanding the predictability of higher or lower than normal volumes during the basin's peak flow period. Results for low and high flows are very similar, which could be related to the high hindcast reliability, and will be described jointly below.

For all target periods/periods of interest, the ROC AUC increases with later initialization dates, reaching a peak with hindcasts generated on May $1^{st}$/June $1^{st}$. This suggests that the potential usefulness of these hindcasts increases as lead time decreases. It is however not the case for hindcasts for July $1^{st}$ to September $30^{th}$, where the ROC AUC decreases after June $1^{st}$. This hints again at a shift from snow to rain as the dominant driver of streamflow between the spring and the summer/fall months.





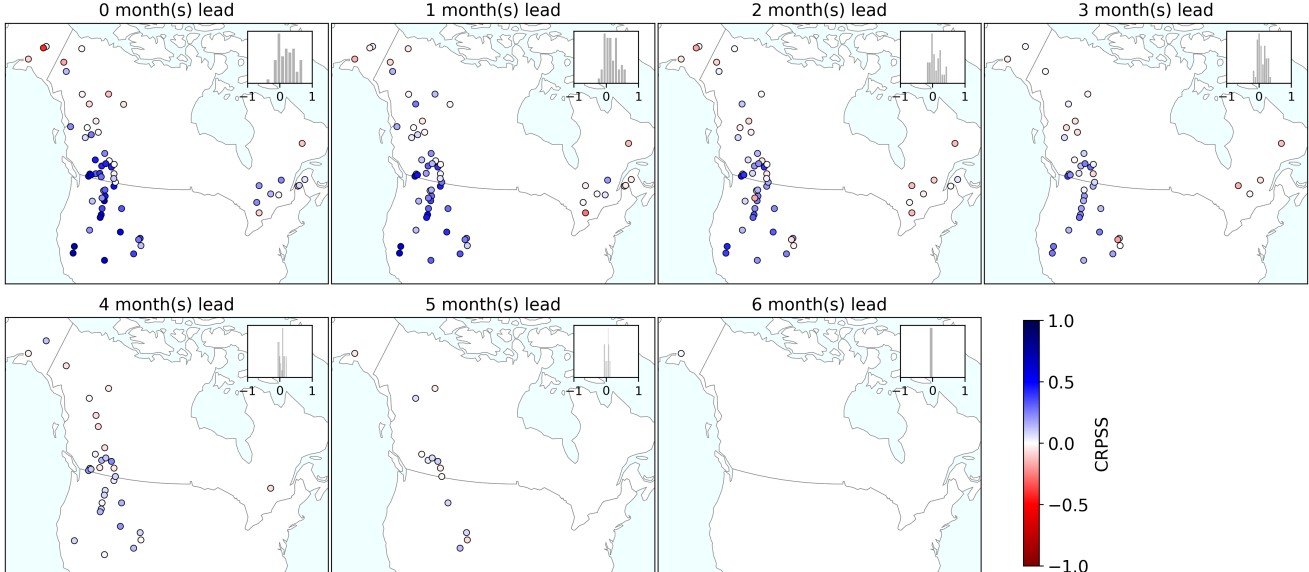

**Figure 7.** Maps of the hindcast CRPSS for hindcasts from zero to six months lead time and inset histograms showing the distribution of CRPSS values. Each subplot shows results for the target period of interest within each river basin. The number of river basins shown on each map varies based on the lead time, reflecting the period of interest being forecasted (e.g., a river basin with a January $1^{st}$ to September $30^{th}$ period of interest cannot be forecasted with more than zero months lead time, after the $1^{st}$ of January). The last subplot (i.e., six months lead time) shows results for a single river basin situated in Alaska.

The hindcast potential usefulness varies for different target periods and the hindcasts generated for the target periods March $1^{st}$ to September $30^{th}$ and May $1^{st}$ to September $30^{th}$ show the best performances, indicating better predictions for low and high flows during these periods. Conversely, the hindcasts produced for the target periods April $1^{st}$ to September $30^{th}$, June 340 $1^{st}$ to September $30^{th}$, and July $1^{st}$ to September $30^{th}$ exhibit the worst performances, implying less predictability for low and high flows for these specific periods. Overall, for all periods of interest except for July $1^{st}$ to September $30^{th}$ there is potential usefulness in predicting low and high flows from January $1^{st}$.

As these plots show results for each basin's period of interest only, some of the boxplots have limited data points (e.g., the boxplots for April $1^{st}$ to September $30^{th}$ are created from results for nine river basins only). This could explain some of the 345 differences between boxplot span. Figure A5 in the Appendix shows results for individual basins.



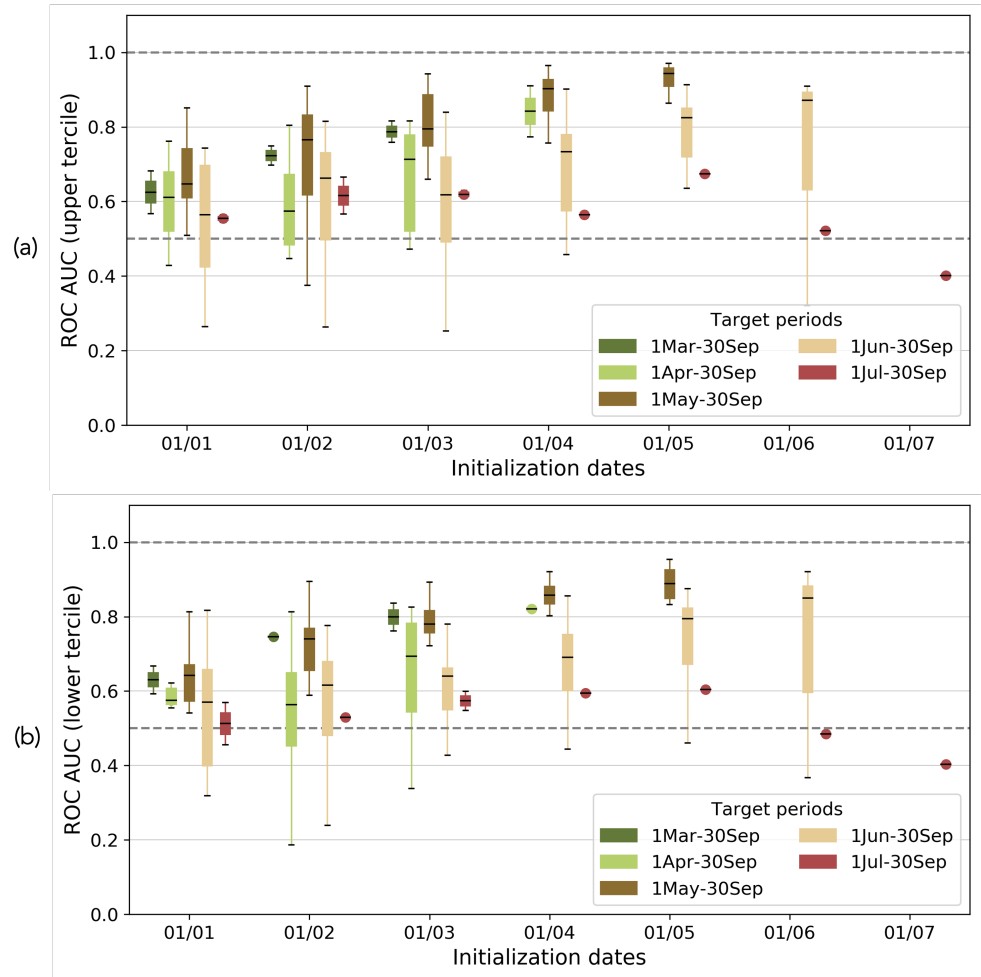

**Figure 8.** Hindcast ROC AUC for each target period as a function of hindcast initialization dates for (a) flows above the climatology upper tercile and (b) flows below the climatology lower tercile. The boxplots display values for all basins, where their period of interest coincides with one of the target periods. The upper dashed line (ROC AUC=1) represents the perfect value and the lower dashed line (ROC AUC=0.5) represents the threshold below which hindcasts have no skill.

# 4  User-oriented discussion

We now draw insights relevant for snow monitoring experts, streamflow forecasters, decision-makers, and workflow developers from the results presented in Section 3.





## 4.1 Snow monitoring experts

For this discussion, snow monitoring experts include snow surveyors, field collection technicians, and monitoring network designers. Collectively, they conduct valuable work to support many different scientific and applied questions. An important use of snow surveys is water supply outlooks. As such, it is worth considering the following questions:

– Which SWE measurement dates are most important for forecasting streamflow volumes?

– Where and when are more SWE data needed for improving streamflow forecasts?

The first question relates to our hypothesis that the highest performances can be found for hindcasts initialized around the peak SWE date in each basin. While peak SWE typically occurs around April $1^{st}$ across North America (Fig. 9 (a)), the results presented in Section 3 reveal that high performance in streamflow forecasts can still be achieved up until June $1^{st}$. This suggests that persistent snowpack (i.e., after April $1^{st}$) can hold important predictability for spring/summer streamflow volumes. Thus, the SWE measurement dates after peak SWE are critical for skilful predictions of streamflow.

The importance of SWE measurement dates depends on station elevation (and possibly also latitude; not shown). As seen with the boxplots in Fig. 9 (b), the timing of peak SWE exhibits a noticeable variation with station elevation, where, in general, stations situated at higher elevations have later peak SWE. On average, stations with peak SWE on February $1^{st}$ and March $1^{st}$ are at lower elevations than stations with peak SWE on April $1^{st}$, May $1^{st}$, and June $1^{st}$. It is evident in the accompanying line histogram in Fig. 9 (b) that the majority of SWE stations are concentrated at lower elevations. While snow depth and

SWE generally increase with elevation, maximum snow depth in mountainous areas typically occurs near the tree line, with some variability across different sites due to variations in canopy cover (Cartwright et al., 2020; Grünewald et al., 2014). This suggests that SWE measurements at mid- to high-elevations best capture peak SWE in these basins.

This brings us to the second question: where and when are more SWE data needed for improving streamflow forecasts? Given the importance of mid- to high-elevation SWE and the limited measurements at these elevations, measurement dates

later in the snow season are necessary to capture the timing and magnitude of maximum SWE and the evolution of snowmelt to predict snowmelt-driven runoff. Investigating the use of snow pillows, snow scales, and snow depth sensors is recommended to provide continuous depth and SWE measurements at point-based survey sites, thus increasing SWE temporal coverage. Expanding spatial coverage of point-based surveys to include more mid- to high-elevation areas may pose challenges due to the difficulty of reaching these locations and the manual labour needed to set up and maintain such sites. This work can hopefully

serve as a guide to getting maximum useful data out of limited observation networks and budgets. Exploring ways to augment SWE spatial coverage may additionally involve replacing point-based SWE data with alternative sources such as remote sensing techniques like Lidar (Painter et al., 2016) or leveraging gridded snow products like SnowCast (http://www.snowcast.ca; Vionnet et al., 2021a) (Mortimer et al., 2020).

This discussion further leads to questions about how additional SWE measurements can improve the quality of the stream-

flow forecasts. There are instances where manual SWE measurements fail to accurately capture the peak snowpack (particularly problematic in the absence of automatic snow measurements within the basin). On top of this, the forecasting strategy utilized



here (i.e., initializing forecasts on the first of each month) may also miss the peak snowpack. To address this, exploring more
frequent predictions, like initializing and updating forecasts in the middle of each month, could prove beneficial. Additional
research aimed at enhancing snow surveying networks could concentrate on identifying station locations that are representative
of the basin SWE at different dates. These specific sites could then be targeted for additional point measurements or continuous
monitoring using snow pillows, snow scales, or snow depth sensors to ensure the comprehensive capture of mid-month peak
SWE events.

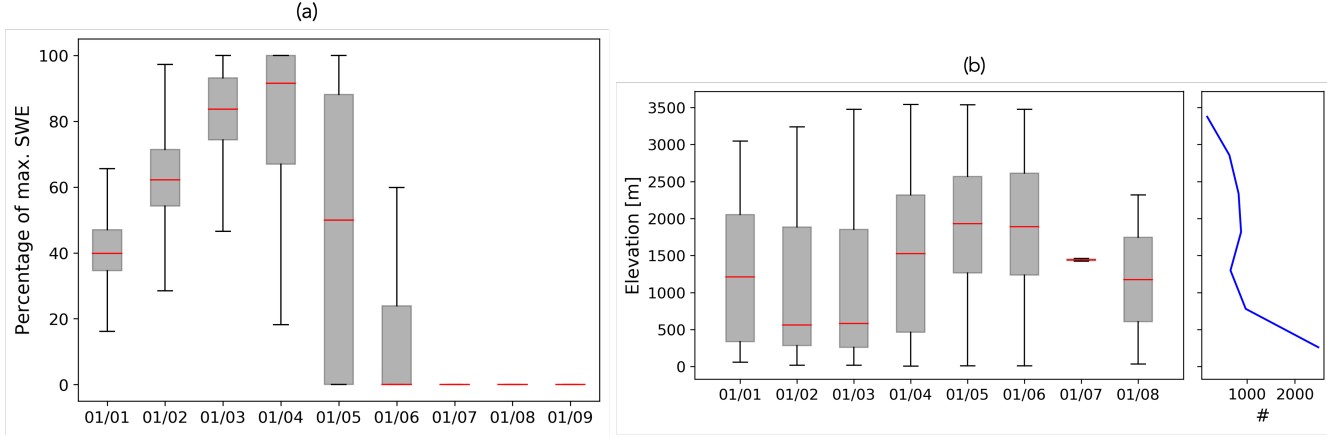

**Figure 9.** (a) SWE content on the first day of each month between January $1^{st}$ and September $1^{st}$. (b) SWE station elevations as a function
of maximum SWE dates and line histogram of the elevation of SWE stations.

## 4.2 Forecasters

According to our second hypothesis, we expect higher hindcast quality for hindcasts with high antecedent SWE content and
low precipitation during the hindcast target period in each basin. We quantify the impact of antecedent snowpack vs. future
precipitation on the hindcast performance. Figure 10 shows the P/SWE ratio (see Section 2.2.5 for more information on its
calculation) as a function of hindcast CRPSS for each target period. As anticipated, for most target periods, the hindcast skill
increases as the P/SWE ratio decreases. This suggests that the hindcasts are more skilful when the initialization date snowpack
increases and/or when the target period precipitation proportion decreases. However, this relationship varies across different
target periods. There is a stronger correlation (with statistical significance) for hindcasts generated for the target periods April
$1^{st}$ to September $30^{th}$ and May $1^{st}$ to September $30^{th}$. Furthermore, this relationship does not hold for target periods August
$1^{st}$ to September $30^{th}$ and September $1^{st}$ to $30^{th}$.

This figure emphasizes the sensitivity of the results to the flow regime, raising concerns about potential loss of predictability
with a changing (snow) climate. According to the Sixth Assessment Report of the Intergovernmental Panel on Climate Change
(IPCC, 2021, 2022), snow extent, snow cover duration, and accumulated snowpack are virtually certain to decline in subarctic
regions of North America. There is also a projected decrease in seasonal snow cover extent and mass in mid- to high-latitudes







**Figure 10.** P/SWE as a function of hindcast CRPSS for all hindcast initialization dates and all basins for each target period (subplots). Each point represents the result for a given basin and initialization date. The Pearson correlation coefficient (r) is shown on the top right of each plot, where the asterisk denotes statistical significance at a 5% significance level.

and at high elevations. Hale et al. (2023) report that the annual snow storage has decreased in over 25% of mountainous areas in western North America between 1950 and 2013, as a result of earlier snowmelt and rainfall in spring months, and declines in winter precipitation.

These changes are predicted to result in snow-related hydrological changes, including declines in snowmelt runoff (except in glacier-fed river basins where the opposite might be true on shorter timescales) (IPCC, 2021, 2022), more frequent rain-





on-snow events at higher elevations (where seasonal snow cover persists) due to a shift from snowfall to rain (Musselman et al., 2018), and consecutive snow drought years in western North America. Berghuijs et al. (2014) show that a change in the precipitation phase from snow to rain significantly decreases the mean streamflow within individual catchments of

the contiguous USA. In snow-dominated regions globally, there is high confidence that peak flows associated with spring snowmelt will occur earlier in the year. This effect has already been documented by several studies that show that new record peak flows fall into time periods outside the nival window (Gillett et al., 2022). Burn and Whitfield (2023) additionally discuss an increasing frequency of rainfall-driven peaks and floods.

As a result of these changes, we expect that the relationships between SWE and streamflow will be affected, impacting the

quality of snow-based streamflow forecasts in the future. Pagano et al. (2004) found that increasing hydrological variability in the western USA was partly responsible for the decline in water supply forecast skill.

The analysis presented here showcases a wide spectrum of predictability, where basins encompass diverse geographies and climates, ranging from purely nival regimes to mixed regimes. This spectrum of predictability can be appreciated in more depth in Fig. A4 in the Appendix. This offers a glimpse into the potential changes in predictability we may observe in the future. To

tackle these questions more thoroughly, future research could look at the impact of snow climate on these results. Additionally, investigating different cross-validation approaches could be influential in maintaining forecast quality over time.

In the study by Zheng et al. (2018), inner mountain areas in the western USA, dominated by snowmelt contribution, showed longer streamflow predictability, while coastal areas, dominated by rainfall contribution, had shorter streamflow predictability. While we sub-selected river basins with a nival regime, we do also notice the influence of future unknown rainfall on these

results. Figure 7 illustrates higher and longer predictability in interior and western North American river basins, contrasting with lower and shorter predictability in the east.

Climate predictors can add to the seasonal streamflow forecast skill available from SWE, especially for basins with strong teleconnections between large-scale climate and local meteorology on longer timescales (Wood et al., 2016; Mendoza et al., 2017). Furthermore, Slater and Villarini (2017) found precipitation variability to be crucial for modeling high flows, while

antecedent wetness impacts low and median flows in the Midwestern USA. They also found that temperature enhances model fits during snowmelt or high evapotranspiration seasons. Lehner et al. (2017) found that the addition of temperature forecast information to operational seasonal streamflow predictions in snowmelt-driven basins within the southwest USA not only enhances the skill of streamflow forecasts but also contributes to mitigating errors in streamflow predictions caused by climate nonstationarity. Antecedent streamflow can also be a strong predictor of future streamflow, as shown by Veiga et al. (2014).

These variables might enhance predictability and warrant exploration. However, the predictability sources vary depending on the initialization date, predictand, basin location, and hydroclimatic features (Wood et al., 2016). Even within a small domain, the relative importance of predictors can differ (Mendoza et al., 2017), emphasizing the need for detailed analysis to put forward additional basin specific predictors. Tools like PyForecast (https://github.com/usbr/PyForecast) could aid in exploring additional predictors for accurate ensemble seasonal volume forecasts within specific river basins or regions using

the workflow presented here.




Additionally, embracing more flexible yet physically accurate forecasting methods is a logical progression. Hybrid methods that combine the strengths of machine learning with process-based models grounded in our comprehension of physical processes emerge as a reasonable choice for enhancing predictability over longer timescales (Slater et al., 2023). In recent studies, Chang et al. (2023) demonstrated the extended predictability of subseasonal hydrological forecasts in Switzerland by

incorporating large-scale atmospheric circulation information. Additionally, Hauswirth et al. (2023) introduced a flexible and efficient hybrid framework that utilized global seasonal forecasts as inputs to produce skilful location-specific seasonal forecast information.

## 4.3 Decision-makers

Forecast reliability plays a crucial role in facilitating risk-based decision-making (Zhao et al., 2016; Mendoza et al., 2017), for

example for determining optimal water release volumes and schedules for hydropower generation and irrigation, or for issuing timely warnings of potential high or low flows. High forecast reliability in turn instills trust in the forecasts for informed decision-making (note that reliability is only one of many factors that contribute to users' trust). Insights from the analysis of a serious game conducted by Crochemore et al. (2021) underscore the importance of high reliability for decision-making. Notably, the study revealed that decision-makers considered high reliability crucial especially for risk-based decision-making

in extreme years.

One of the distinguishing strengths of statistical forecasts, such as the ones generated here, over process-based forecasts lies in their ability to achieve high forecast reliability, stemming from the ensemble generation method employed. This aligns with the findings of Mendoza et al. (2017), who found that the regression-based forecasting methods they examined exhibited higher reliability than the process-based forecasting methods. Emerton et al. (2018) found that the process-based seasonal streamflow

forecasts produced within the Global Flood Awareness System (GloFAS) had limited reliability globally, with some spatial variability.

A fundamental question that arises pertains to the temporal horizon within which decisions can be confidently made. To provide some initial insights for decision-making, we provide matrices showing the evolution of the forecasts' potential usefulness for predicting low and high flows within specific river basins with increasing lead time, considering the period of interest

within each basin (Fig. A5 in the Appendix). However, the answer to this question is inherently user-specific, and depends on factors such as the choice of baseline, target periods, and specific events or thresholds of interest. To address these specific user requirements, further analysis is essential. This can be achieved by building on the provided codes, and tailoring the forecasting methodology to align with distinct user needs.

In the context of operational forecasting, forecast consistency is a critical aspect to ensure coherent decisions throughout

the decision-making period. Considering the findings presented in this paper from an operational forecasting standpoint, a few methodological decisions may have affected the results. In this analysis, we conducted a PCA and established new models for each predictor-predictand combination and each year left out. We adopted a leave-one-out cross-validation approach due to limited data in certain basins, leveraging all available data to generate new hindcasts. It is important to acknowledge that this approach might introduce inconsistency from month to month and year to year (Garen, David C., 1992), as well as some



artificial quality in the hindcast verification process (DelSole and Shukla, 2009). In operational scenarios, forecasters may opt to use pre-existing PC matrices and models to ensure forecast consistency and ensure smooth decision-making. However, this could be problematic in case of non-stationary input data (Shen et al., 2022). This topic warrants further attention.

## 4.4 Workflow developers

Reproducibility of research in the water sciences is still very low (Stagge et al., 2019). This contributes to the typically slow
transfer of research to operations. While journal policies are moving towards more open science (e.g., Blöschl et al., 2014; Clark et al., 2021a), such policies are not yet at the stage where full workflows must be published alongside a paper - though this seems a logical next step.

Building workflows that are both intuitive (i.e., that can represent our understanding of local hydro-meteorological processes; (Veiga et al., 2014)) and reproducible is essential to providing platforms for progressive and purposeful testing of new scientific
advances, and to pave the way for applying research outcomes in practice. Furthermore, it fosters more equitable water research and education (Castronova et al., 2023).

However, it is important to acknowledge that the demands of scientific journals for open-source data and methods may sometimes conflict with the rapid and competitive nature of some environments, including academia. Striking a balance between open collaboration and maintaining a competitive edge poses challenges that the academic community must address. Explicitly
acknowledging a researcher's commitment to transparency, reproducibility, and reusability of their work during merit reviews is one possible step forward.

The workflow developed as part of this study adheres to the principles of open and collaborative science, facilitated by its design (i.e., Jupyter Notebooks) and code-sharing (i.e, GitHub). In line with the recommendations by Knoben et al. (2022), our approach prioritizes clarity, modularity, and traceability in the workflow design. This enables users to easily adapt the workflow
for any river basin in the USGS or the WSC HYDAT datasets. Users have the flexibility to modify, enhance, or replace specific components of the workflow to suit their needs. Below is a non-exhaustive list of future research ideas.

- – In Notebook 1, one could look into replacing the regime classification component of this workflow with an alternative method to identify basins with a nival regime (such as using the fraction of precipitation falling as snow).

- – In Notebook 2, we set the end of the water year as the endpoint for all forecast target periods in all river basins. Yet, some
of these river basins may experience late summer to early fall rainfall events. For example, river basins in the east which can be impacted by extratropical storms during that time of year and show a mixed hydrological regime (Burn et al., 2016). While we discarded most of these river basins through the strict regime classification/basin selection, it could be that some of these river basins were retained, affecting the forecast quality.

- – In Notebook 3,

- – future research could explore the impact of using different gap filling methods. An example is the various gap filling strategies explored by Tang et al. (2020a) for meteorological stations infilling to create the SCDNA dataset.





– We used the SCDNA precipitation data for infilling, which does not distinguish between solid and liquid precipitation. Additionally, the precipitation was accumulated during the entire water year and did not consider the onset of snowmelt. Both of these decisions could have led to lower correlations between the SWE and the cumulative precipitation to identify suitable donor stations.


– The selection of SWE stations used as predictors could play a significant role in the forecast quality. To improve SWE sampling, future research may consider expanding the station selection to include those within a buffer of the basin. Although this method was coded as part of the workflow, it was not implemented in this paper due to the need for a more comprehensive analysis of its impact on forecast quality.

– In Notebook 4,

– there was no established minimum threshold for the percentage of variance that PC1 should explain in order to be used as a predictor. In addition, although the ability to use additional PCs was also coded as part of the workflow, it was not further explored in this paper in order to avoid overfitting. There exist various methodologies around stopping criteria for including predictors, such as the Bayesian information criterion (BIC), or regularization ap-

proaches that can lessen the risk of overfitting (Baker et al., 2020). Investigating the effects of using additional PCs could lead to valuable insights. For instance, it could provide a means to investigate whether this accounts for the consistently underestimated variability.

– Subsequent studies may explore a range of cross-validation strategies (e.g., sample-splitting, increasing the number of omitted years, or excluding extreme years from the training dataset), to assess how they affect the quality of the

generated hindcasts.

## 5 Conclusions

We have developed a systematic and reproducible data-driven workflow for probabilistic seasonal streamflow forecasting in snow-fed river basins across North America, including Canada and the USA. This structured workflow consists of five essential steps: 1) Regime classification and basins selection, 2) Streamflow pre-processing, 3) SWE pre-processing, 4) Forecasting, and

5) Hindcast verification. This methodology was applied to 75 basins characterized by a nival (snowmelt-driven) regime and limited regulation across diverse North American geographies and climates. The input data, spanning from 1979 to 2021, includes SWE (predictor), precipitation (for gap filling), and streamflow (predictand) station data. The ensemble hindcasts were generated monthly, with initialization dates ranging from January $1^{st}$ to September $1^{st}$, and target periods January $1^{st}$ - September $30^{th}$, February $1^{st}$ - September $30^{th}$, and so on. We analyze the hindcasts using deterministic metrics (i.e., the

KGE″ and its decomposition to measure correlation, bias and variability) and probabilistic metrics (i.e., the reliability index, CRPSS, and ROC AUC, to measure reliability, skill and potential usefulness, respectively). The insights derived from this comprehensive analysis are invaluable for snow monitoring experts, forecasters, decision-makers, and workflow developers.

Key findings include:





– **For snow monitoring experts:** Late-season snowpack (i.e., after April $1^{st}$) holds significant predictability for spring/-summer volumes. Thus, capturing snowpack beyond the peak period is crucial for skilful predictions.

– **For forecasters:** Higher hindcast skill is achievable using this forecasting approach for target periods when basins exhibit high antecedent SWE content and low precipitation during the forecast period. In many river basins and times of year, SWE is not a key predictor. Therefore, an optimal approach should leverage climate predictors to achieve a more comprehensive balance between the initial conditions and meteorological forcings that contribute to the predictability of runoff.

– **For decision-makers:** This statistical forecasting approach, not unlike other statistical forecasting approaches, can generate ensemble hindcasts with that are statistically reliability. Moreover, for all periods of interest up to and including June $1^{st}$ to September $30^{th}$, we can predict lower than normal and higher than normal streamflows with up to five month lead time.

– **For workflow developers:** The developed workflow, shared as Jupyter Notebooks on GitHub, follows the principles of open and collaborative science. Its design is clear, modular, traceable, intuitive, and reproducible. This in turn facilitate applications in other cold regions, and the advancement of methods based on the benchmark provided. We invite others to build upon this workflow and have outlined potential improvements in Section 4.4.

This study contributes to the existing research by: 1) expanding the spatial scope to encompass both Canada and the USA, 2) creating a completely open and reproducible workflow, and 3) offering practical guidance for diverse users.

*Code and data availability.* The Python codes used to generate all hindcasts analyzed in this paper are available on Zenodo (Arnal et al., 2023b, v0.9.0). The release additionally contains compiled datasets of the basin shapefiles and the daily streamflow observations used, described in more detail in the associated readme. A user-fiendly version of the FROSTBYTE workflow is available on GitHub (Arnal et al., 2023a, v0.9.0), with sample data for two river basins to support reproducibility.

*Author contributions.* LA designed the workflow with the guidance of MC, AP, VV, VF, PW, DC, and AW. DC contributed to the development of the workflow GitHub repository. MC, VV and WK downloaded parts of the input data to the workflow. LA generated all hindcasts evaluated in this paper. LA prepared the manuscript with contributions from all co-authors.

*Competing interests.* The authors declare that they have no conflict of interest.





*Acknowledgements.* The authors acknowledge funding support from Environment and Climate Change Canada (ECCC), from the Canada
First Research Excellence Fund's Global Water Futures program, and from NOAA CIROH. David Casson was partly funded by Deltares
Strategic Research under the Dutch Subsidy for Institutes for Applied Research. Andrew Wood is supported by Reclamation under Intera-
gency Agreements R21PG00095 and R22PG00035, by the U.S. Army Corps of Engineers Climate Preparedness and Resilience Program,
and by NOAA CIROH under subaward A22-0310-S001-A02.

We would like to specifically thank the following people for their guidance throughout the workflow and manuscript preparation, in no
specific order:

– Khaled Akhtar, Evan Friesenhan, Jennifer Nafziger, and other forecasters from the Government of Alberta in Edmonton, for discussions
on streamflow forecasting and local challenges.

– Alex Cebulski, Dennis Rollag, and Marina Tait for a discussion on snow surveying.

– Pablo Mendoza for discussions around the hindcast evaluation.

– Guoqiang Tang, Kasra Keshavarz, and Shervan Gharari for data and example codes.

– Colleen Mortimer for compiling SWE data for New England (USA) used in this study.

– Alida Thiombiano for a revision of the SWE gap filling script.

We additionally acknowledge the contribution of SWE data by the Ministère de l'Environnement, de la Lutte contre les changements
climatiques, de la Faune et des Parcs (MELCC, 2019).

The authors acknowledge that collectively they reside on traditional territories of the peoples of Treaties 6 and 7, which
include the Cree, the Haudenosaunee, the Ktunaxa, the Mohawk, the Niitsitapi (Blackfoot Confederacy, comprised of the
Siksika, the Piikani, and the Kainai First Nations), the Stoney Nakoda (including Chiniki, Bearspaw, and Goodstoney First
Nations), and the Tsuut'ina First Nation, on the homelands of the Métis, on the ancestral homelands of the Cheyenne, Arapaho,
Ute and many other Native American nations, as well as on unceded Indigenous lands of which the Kanien'kehá:ka Nation are
the custodians. The authors thank these nations for their care and stewardship over this land and water and pay their respect
to the ancestors of these places. It is hoped that this article helps to improve river flow forecasting worldwide to protect all its
communities and people.



## Appendix A: Additional figures

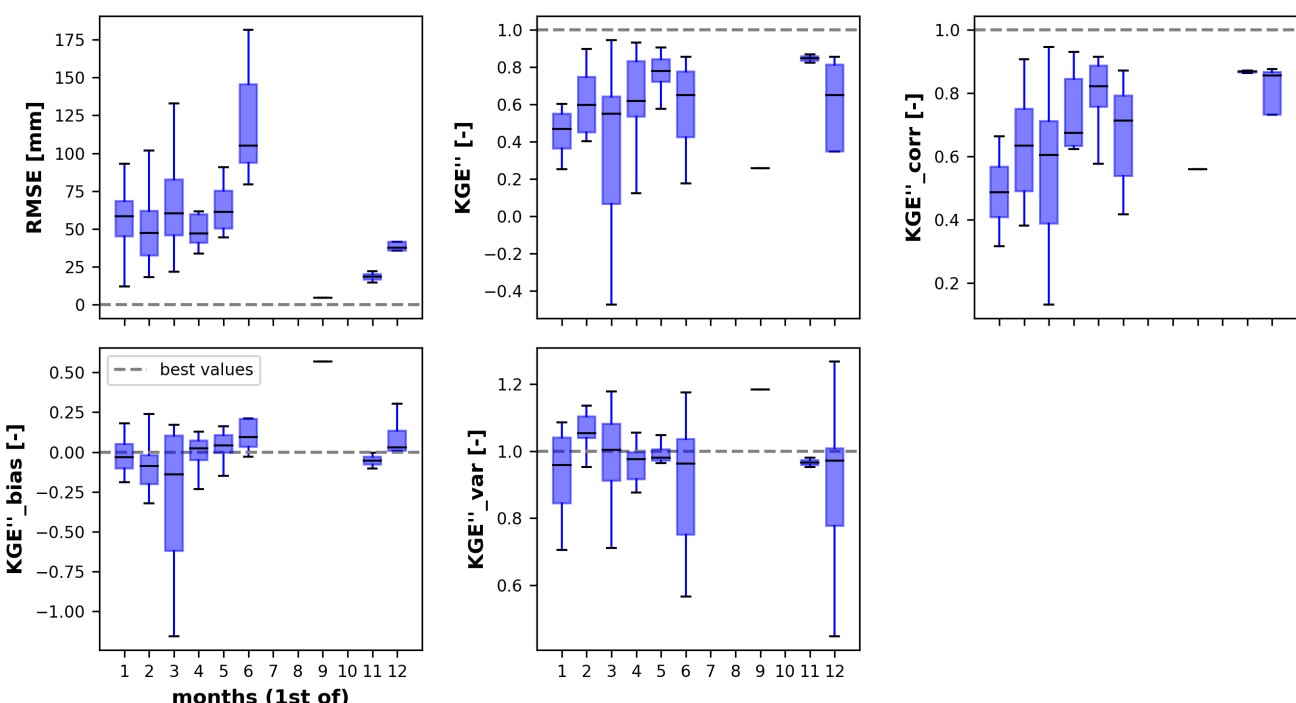

**Figure A1.** Performance metrics obtained from the artificial gap filling step for the Bow River at Banff (Alberta, Canada). The boxplots contain results for all SWE stations within the river basin.



**Figure A2.** Time series of the availability of SWE station data on the first day of each month (subplots) before and after gap filling for the Bow River at Banff (Alberta, Canada).





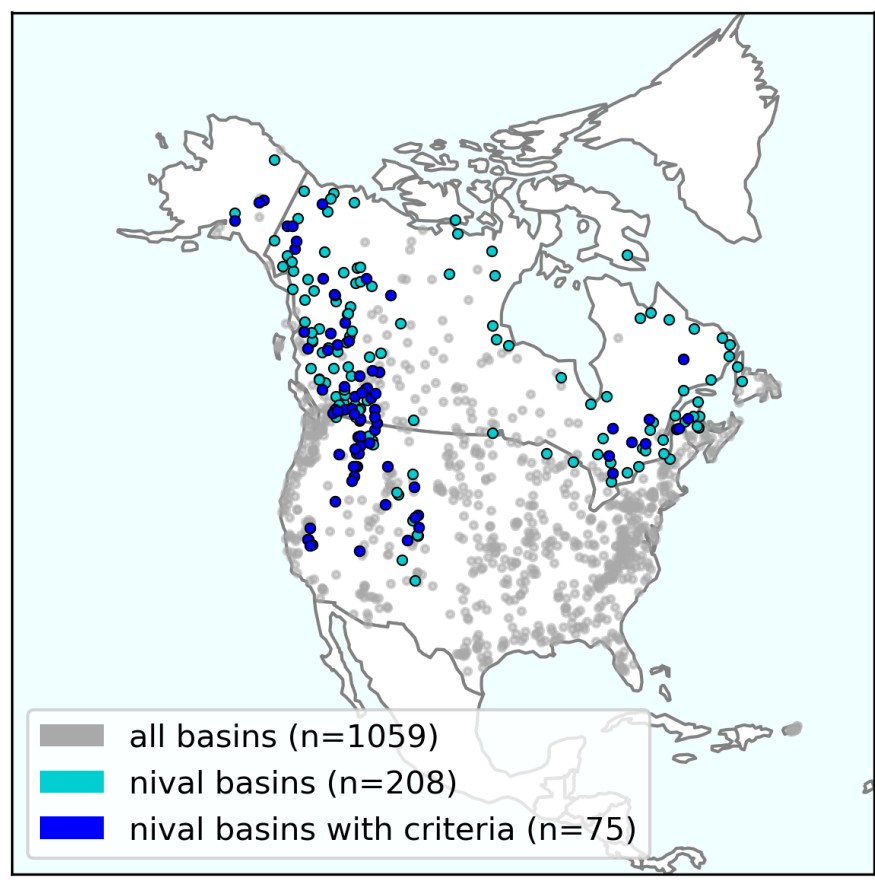

**Figure A3.** Map of all North American basins with data for the period 1979–2021 and with limited regulation (grey), identified nival basins (turquoise), and the subset of nival basins meeting the data requirements for the forecasting analysis presented in this paper (dark blue).





**Figure A4.** Bootstrapping mean and range (5th to 95th percentiles) for various metrics across the selected nival river basins (sorted from lowest to highest mean metric value). Results are shown for June $1^{st}$ to September $30^{th}$ hindcasts generated on June $1^{st}$ for illustrative purposes.





**Figure A5.** Hindcast ROC AUC for each basin as a function of initialization dates (a) for flows below the climatology lower tercile and (b) for flows above the climatology upper tercile (right). Results are shown only for each basin's period of interest. Basins are ordered from North to South, based on their latitudes. Blue colours show potential useful hindcasts and red colours show hindcasts with no skill.



## Appendix B: Circular statistics

The equations used for the regime classification were taken from Burn et al. (2010). The date of occurrence of an event ($i$) is defined by converting the Julian date to an angular value ($\theta_i$; in radians), using the formula:

$$\theta_i = (JulianDate_i)\left(\frac{2\pi}{lenyr}\right) \tag{B1}$$

where $lenyr$ is the number of days in a year.

From a sample of $n$ events, we can find the x- and y-coordinates of the mean date with the following formulas:

$$\bar{x} = \frac{1}{n}\sum_{i=1}^{n} cos\theta_i \tag{B2}$$

$$\bar{y} = \frac{1}{n}\sum_{i=1}^{n} sin\theta_i \tag{B3}$$

The mean date ($MD$), or average date of occurrence of all events $i$, can then be obtained with:

$$MD = tan^{-1}\left(\frac{\bar{y}}{\bar{x}}\right)\left(\frac{lenyr}{2\pi}\right) \tag{B4}$$

Finally, the regularity ($\bar{r}$) of the $n$ event occurrences can be determined with the formula:

$$\bar{r} = \sqrt{\bar{x}^2 + \bar{y}^2} \tag{B5}$$

where $\bar{r}$ is a dimensionless measure of the spread in the dates of occurrences of the $n$ events, which varies from zero to one. Larger values indicate a higher level of regularity.

## Appendix C: KGE'' decomposition

The equations for the components of the KGE'' were taken from Clark et al. (2021b). The bias ratio ($\beta$) is:

$$\beta = \frac{(\mu_s - \mu_o)^2}{\sigma_o^2} \tag{C1}$$

where $\mu_s$ is the mean of the simulations, $\mu_o$ is the mean of the observations, and $\sigma_o^2$ is the variance of the observations. $\beta$ has a perfect score of zero.

The variability ratio ($\alpha$) is:

$$\alpha = \frac{\sigma_s}{\sigma_o} \tag{C2}$$

$\alpha$ has a perfect score of one.

The correlation ($\rho$) is the Pearson correlation between the simulations and the observations, and has a perfect score of one.





## Appendix D: SWE and precipitation statistics

We computed the Snow Water Equivalent (SWE) content historical median for each initialization date $i$ and each SWE station $s$ using the formula:

$$SWEcontent_{i,s} = med\left(\frac{SWE_{i,wy,s}}{max(SWE)_{wy,s}} \times 100\right) \tag{D1}$$

where $SWE_{i,wy,s}$ is the SWE on initialization date $i$ within water year $wy$ and for SWE station $s$, and $max(SWE)_{wy,s}$ is the maximum SWE for water year $wy$ for SWE station $s$.

To determine the precipitation to SWE ratio, we first calculated the basin mean cumulative precipitation historical median for each target period $t$ and each nival basin $b$ with:

$$Pstats_{t,b} = \frac{1}{n}\sum_{s=1}^{n} med(Pcumul_{t,s}) \tag{D2}$$

where $n$ in the total number of precipitation stations $s$ in basin $b$.

Next, we calculated the basin mean SWE historical median for each initialization date $i$ and each nival basin $b$ with:

$$SWEstats_{i,b} = \frac{1}{n}\sum_{s=1}^{n} med(SWE_{i,s}) \tag{D3}$$

where $n$ in the total number of SWE stations $s$ in basin $b$.

Finally, the precipitation to SWE ratio was determined for each combination of initialization date $i$, target period $t$, and nival basin $b$ with:

$$P/SWE_{t,i,b} = \frac{Pstats_{t,b}}{SWEstats_{i,b}} \tag{D4}$$

The precipitation to SWE ratio ranges between $-\infty$ and $+\infty$.





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
