# Peer review of "FROSTBYTE: A reproducible data-driven workflow for probabilistic seasonal streamflow forecasting in snow-fed river basins across North America"

_EGUsphere, 2023_

## Author Comment (AC1)

**FROSTBYTE: A reproducible data-driven workflow for probabilistic seasonal streamflow forecasting in snow-fed river basins across North America**

Response to RC1

In this paper, the authors have presented a data-driven workflow for ensemble seasonal streamflow forecasting using snow water equivalent as predictors. The findings offer valuable insights relevant to various stakeholders, such as forecasters and decision-makers, effectively merging scientific precision with practical workflow development insights. The subject matter is of current interest and contributes insights to hydrological forecasting, both the workflow, and the knowledge of the predictability of streamflow from late-season snowpack. For a deeper comprehension of the study, I propose additional discussions with the authors, detailed below.

We thank the reviewer for their positive and constructive comments on our manuscript. Their comments are copy-pasted below verbatim in black, and our responses are underneath each comment in blue.

Line 169 and 206, could you clarify whether an independent regression model is employed for each target period at every initialization date, or if a single model is capable of generating multiple outputs for all target periods at the same initiation date?

Hindcasts are indeed generated using an independent regression model for each river basin, initialization date, target period, and year left out. This is mentioned in Section 2.2.4, on L224-225 of the initial manuscript: "We conduct a PCA and fit a new model for each predictor-predictand combination".

We have clarified this point in the text: "An independent regression model is used to produce an ensemble hindcast for each river basin, initialization date, target period, and year left out".

Line 221, could you provide the information of the average explained variance of the PCs?

We agree that this would be useful information for the readers and will add this information in the revised manuscript.

Line 293, would be interested to know what are general reliability of other methods, to better understand how much improvement the proposed method obtains.

Comparing our reliability results to the reliability of other methods is not straightforward given the variety of approaches used to evaluate reliability in the literature. However, our results could be compared to those from Mendoza et al. (2017), who compared the reliability index of statistical, process-based and hybrid methods for seasonal streamflow

forecasting at five case study sites across the USA Pacific Northwest region. They found that, overall, the statistical methods yielded the most reliable hindcasts.

We have added some more detail about the reliability index values obtained with the various methods in the revised manuscript: "For five case study sites across the USA Pacific Northwest, their regression-based methods achieved reliability index values ranging between 0.6 and nearly 1, while the reliability of the process-based ensemble streamflow prediction (ESP) hindcasts declined when approaching the April 1st initialization date, with an overall reliability index ranging between 0.4 and 0.9. Our approach yielded reliability index values comparable to those obtained from the statistical methods developed by Mendoza et al. (2017)".

Line 300, please considering to rescale the y axis to make the difference more visible.

We have changed the figure's y-axis limits so that results are clearer - thank you.

Line 310, here the authors mentioned the limitation of comparing between systems with different ensemble members. Would it be more comparable to use fairCRPSS instead?

We thank the reviewer for this idea and will calculate the fair-CRPSS as well to compare results against the standard CRPSS, prior to adding these results to the manuscript.

Line 321, two "and" here.

Thank you for catching this - we have corrected this in the revised manuscript.

Line 323, please specify which basins are the ones that display low to no skill throughout all initialization dates. And does this refer back to Fig.6 since there is no initialization date information in Fig.7.

As described in the text, basins with low to no skill are situated in the northwest and in the east. We will however add a few basin names to add more in-depth information to the manuscript.

We have additionally removed the sentence "On the other hand, some basins display low to no skill throughout all initialization dates" on L323-324, as this was a repetition of the first sentence of this paragraph on L319-320. Further, as pointed out by the reviewer, initialization dates are not shown on Fig. 7 (at least not explicitly).

We now also provide some clarifying information on the lead months displayed on Fig. 7 in the revised manuscript, in both the text and the caption respectively: "Note that the lead months are different from the initialization dates of the hindcast, where lead month refers to the number of months between the hindcast initialization date and the target period start", and "Note that initialization dates and periods of interest are not shown

explicitly here. For instance, the first map, showing the CRPSS for hindcasts with zero months lead time, will include results from hindcasts of January 1st to September 30th initialized on January 1st, as well as from hindcasts of February 1st to September 30th initialized on February 1st, etc. On the other hand, the last map, showing the CRPSS for hindcasts with six months lead time, will include results from hindcasts of July 1st to September 30th or later, initialized on January 1st or later".

Line 332, an additional interesting pattern from this figure is that the peak skill for each target period appears when the initialization month is at the start of the target period, e.g. the target period of April to September, the peak occurs when initialized at April 1st.

This is indeed a noteworthy finding which we mention, albeit differently, on L332-334: "For all target periods/periods of interest, the ROC AUC increases with later initialization dates, reaching a peak with hindcasts generated on May 1st/June 1st. This suggests that the potential usefulness of these hindcasts increases as lead time decreases. It is however not the case for hindcasts for July 1st to September 30th, where the ROC AUC decreases after June 1st."

We have rephrased so it is clearer and the revised text now reads: "For most target periods/periods of interest, peak ROC AUC is obtained for hindcasts with zero months lead time. For example, the ROC AUC of hindcasts for May 1st to September 30th is the highest when the hindcasts are initialized on May 1st. The ROC AUC, and therefore the potential usefulness, of most hindcasts decreases with increasing lead time. This is however not the case for hindcasts for July 1st to September 30th, where the ROC AUC is highest when hindcasts are initialized on average on May 1st (with two months lead time)".

---

## Author Response (AR1)

**FROSTBYTE: A reproducible data-driven workflow for probabilistic seasonal streamflow forecasting in snow-fed river basins across North America**

Response to RC1

In this paper, the authors have presented a data-driven workflow for ensemble seasonal streamflow forecasting using snow water equivalent as predictors. The findings offer valuable insights relevant to various stakeholders, such as forecasters and decision-makers, effectively merging scientific precision with practical workflow development insights. The subject matter is of current interest and contributes insights to hydrological forecasting, both the workflow, and the knowledge of the predictability of streamflow from late-season snowpack. For a deeper comprehension of the study, I propose additional discussions with the authors, detailed below.

We thank the reviewer for their positive and constructive comments on our manuscript. Their comments are copy-pasted below verbatim in black, and our responses are underneath each comment in blue. All line numbers refer to the marked-up manuscript version, unless specified otherwise.

Line 169 and 206, could you clarify whether an independent regression model is employed for each target period at every initialization date, or if a single model is capable of generating multiple outputs for all target periods at the same initiation date?

Hindcasts are indeed generated using an independent regression model for each river basin, initialization date, target period, and year left out. This is mentioned in Section 2.2.4, on L224-225 of the original manuscript: "We conduct a PCA and fit a new model for each predictor-predictand combination".

We have clarified this point in the text (L280-281): "An independent regression model is used to produce an ensemble hindcast for each river basin, initialization date, target period, and year left out".

Line 221, could you provide the information of the average explained variance of the PCs?

We agree that this would be useful information for the readers and will add this information in the revised manuscript. We found that the first principal component explains 90% of the total variance in the gap filled SWE stations dataset.

We added a sentence to provide this information in the manuscript (L264-266): "In our analysis, the first PC explains 90% of the total variance in the gap filled SWE stations dataset, on average across all hindcast initialization dates and river basins. The explained variance of each principal component can be found in the Appendix (Fig. A3)". We also made a plot showing the explained variance of each principal component which we added to the manuscript Appendix.

Line 293, would be interested to know what are general reliability of other methods, to better understand how much improvement the proposed method obtains.

Comparing our reliability results to the reliability of other methods is not straightforward given the variety of approaches used to evaluate reliability in the literature. However, our results could be compared to those from Mendoza et al. (2017), who compared the reliability index of statistical, process-based and hybrid methods for seasonal streamflow forecasting at five case study sites across the USA Pacific Northwest region. They found that, overall, the statistical methods yielded the most reliable hindcasts.

We have added some more detail about the reliability index values obtained with the various methods in the revised manuscript (L542-546): "For five case study sites across the USA Pacific Northwest, their regression-based methods achieved reliability index values ranging between 0.6 and nearly 1, while the reliability of the process-based ensemble streamflow prediction (ESP) hindcasts declined when approaching the April 1st initialization date, with an overall reliability index ranging between 0.4 and 0.9. Our approach yielded reliability index values comparable to those obtained from the statistical methods developed by Mendoza et al. (2017)".

Line 300, please considering to rescale the y axis to make the difference more visible.

We have changed Fig. 5's y-axis limits so that results are clearer - thank you.

Line 310, here the authors mentioned the limitation of comparing between systems with different ensemble members. Would it be more comparable to use fairCRPSS instead?

We thank the reviewer for this idea. We have calculated the Fair CRPSS and compared the results against those obtained with the standard CRPSS. While the key messages remain unchanged, we have decided to replace the CRPSS results with the Fair CRPSS results in the manuscript to ensure scientific robustness.

Here is a list of the content we changed in the manuscript:
- We have replaced the boxplots (Fig. 6), maps (Fig. 7), the CRPSS vs. SWE/P scatterplots (Fig. 10) and the appendix figure of verification results sorted by mean value (Fig. A7) with the new results based on the Fair CRPSS.
- We have replaced all mentions of the "CRPSS" in the text with the "Fair CRPSS" instead, where relevant.
- We rewrote the methods regarding the CRPSS calculation (Table 1).

Line 321, two "and" here.

Thank you for catching this - we have corrected this in the revised manuscript.

Line 323, please specify which basins are the ones that display low to no skill throughout all initialization dates. And does this refer back to Fig.6 since there is no initialization date information in Fig.7.

As described in the text, basins with low to no skill are situated in the northwest and in the east.

We have now added a figure in the Appendix (Fig. A5) displaying river basins with no or negative skill and those with high skill consistently throughout all lead times for the basin's target period of interest. Note that these results are also based on the Fair CRPSS. We have also added a sentence in the results section to refer to this new figure (L380-381): "Figure A5 in the Appendix displays river basins which consistently exhibit negative skill, as well as those consistently demonstrating high skill".

We have additionally removed the sentence "On the other hand, some basins display low to no skill throughout all initialization dates" on L381-382, as this was a repetition of the first sentence of this paragraph. Further, as pointed out by the reviewer, initialization dates are not shown in Fig. 7 (at least not explicitly).

We now also provide some clarifying information on the lead months displayed in Fig. 7 in the revised manuscript, in both the text (L370-372) and Fig. 7's caption respectively: "Note that the lead months are different from the initialization dates of the hindcast, where lead month refers to the number of months between the hindcast initialization date and the target period start", and "Note that initialization dates and periods of interest are not shown explicitly here. For instance, the first map, showing the CRPSS for hindcasts with zero months lead time, will include results from hindcasts of January 1$^{st}$ to September 30$^{th}$ initialized on January 1$^{st}$, as well as from hindcasts of February 1$^{st}$ to September 30$^{th}$ initialized on February 1$^{st}$, etc. On the other hand, the last map, showing the CRPSS for hindcasts with six months lead time, will include results from hindcasts of July 1$^{st}$ to September 30$^{th}$ or later, initialized on January 1$^{st}$ or later".

Line 332, an additional interesting pattern from this figure is that the peak skill for each target period appears when the initialization month is at the start of the target period, e.g. the target period of April to September, the peak occurs when initialized at April 1st.

This is indeed a noteworthy finding which we mention, albeit differently, on L389-391 of the original manuscript: "For all target periods/periods of interest, the ROC AUC increases with later initialization dates, reaching a peak with hindcasts generated on May 1$^{st}$/June 1$^{st}$. This suggests that the potential usefulness of these hindcasts increases as lead time decreases. It is however not the case for hindcasts for July 1$^{st}$ to September 30$^{th}$, where the ROC AUC decreases after June 1$^{st}$".

We have rephrased so it is clearer and the revised text now reads (L397-401): "For most target periods of interest, peak ROC AUC is obtained for hindcasts with zero months lead

time (higher ROC AUC is better). For example, the ROC AUC of hindcasts for May 1$^{st}$ to September 30$^{th}$ is the highest when the hindcasts are initialized on May 1$^{st}$. The ROC AUC, and therefore the potential usefulness, of most hindcasts decreases with increasing lead time. This is however not the case for hindcasts for July 1$^{st}$ to September 30$^{th}$, where the ROC AUC is highest when hindcasts are initialized on average on May 1$^{st}$ (with two months lead time)".

**FROSTBYTE: A reproducible data-driven workflow for probabilistic seasonal streamflow forecasting in snow-fed river basins across North America**

Response to RC2

The manuscript by Louise Arnal et al. presents a new data-driven workflow for probabilistic seasonal streamflow forecasting in North America, based on snow water equivalent (SWE) as the sole predictor for streamflow forecasting. A special emphasize of the work is put on the reproducibility of the workflow, which can not only be found in the elaborate description of the workflow and the graphical methods but also the collection of open source Jupyter Notebooks of every workflow step. The probabilistic forecasting system was used to create ensemble hindcasts for different target periods, which are relevant for different users such a s snow monitoring experts, forecasters and decision-makers, and were assessed using deterministic and probabilistic metrics. The discussion reviewed relevant insights and findings from the analysis in a refreshing setup, focusing again on the specific users, giving suggestions for future improvements and opportunities to utilize the presented workflow as well as offering practical guidance. Overall, this work does not only present a promising probabilistic forecasting system for local streamflow forecasting in snow-fed river basins, a well-documented workflow, that creates the opportunity for easy implementation for end users, but also is a great example on how research can be presented in a transparent and thorough manner, following principles of open and collaborative science.

We thank the reviewer for their positive and constructive comments on our manuscript. Their comments are copy-pasted below verbatim in black, and our responses are underneath each comment in blue. All line numbers refer to the marked-up manuscript version, unless specified otherwise.

The following points, remarks and questions are mostly raised for further clarifications, no major comments.

Minor comments:

Section 2.1.1, line 81: could you elaborate how catchments with 'limited regulations' are defined? Are there specific or more general criteria that label catchments as 'regulated'? and how does it vary for the different catchments throughout North America?

In the USA, stations included in HCDN-2009 meet the following criteria: (1) they are identified as being in current "reference" condition according to the GAGES-II classification; (2) they have at least 20 years of complete and continuous discharge record through water year 2009; (3) they have less than 5% impervious surface area; (4) they were not eliminated during a review by participating State USGS Water Science Centers (Lins, 2012).

The GAGES-II reference sites were defined based on the following criteria: (1) hydrological disturbance of the watershed (measured with an index taking into account geospatial measures of reservoir storage, dam locations and density, freshwater withdrawal, road density, and the U.S. Environmental Protection Agency's National Pollutant Discharge Elimination System (NPDES) discharges) was less than 75% of all other gauged watershed in its region; (2) the USGS Annual Water Data Reports did not identify the presence of "regulated" streamflows; (3) the watershed passed a visual screening using satellite imagery for the presence of human activities that suggested flow diversion, groundwater withdrawal, and other factors known to influence natural streamflows (Lins, 2012).

In Canada, the requirements for an RHBN station are: (1) minimum of 20 years of data with few small gaps, and a preference for full-year data instead of seasonal only; (2) minimal or stable human impacts on the watershed as defined by the agricultural and urban lands, road density, population density, presence and significance of flow structures. The RHBN subset, RHBN-N, used in this study, comprises 318 stations and was created to represent a nationally balanced network that represents the best available stations among similar watersheds (ECCC, 2021).

We have added a summary of these criteria in the revised manuscript, and we now also highlight the fact that the selection criteria are not identical between the two datasets. The revised text is (L85-98): "For the USA, we use shapefiles and streamflow observations for basins with limited regulation from the USGS Hydro-Climatic Data Network 2009 (HCDN-2009; Lins, 2012; Falcone, 2011). HCDN-2009 comprises stations with minimal hydrological disturbance, measured by the presence of dams, freshwater withdrawal, including from groundwater, flow diversion, roads and other impervious surface areas, and pollutant discharges. Moreover, inclusion in the dataset necessitated a minimum of 20 years of continuous availability of streamflow data. For Canada, we use shapefiles and streamflow observations for basins with limited regulation from the Water Survey of Canada (WSC) HYDAT Reference Hydrometric Basin Network (RHBN) subset, called RHBN-N (ECCC, 2021). The selection criteria for the HCDN-2009 and RHBN datasets exhibit substantial similarity, albeit with potential methodological nuances that may stem from varying priorities and contexts. The reference hydrologic networks include only stations considered to have minimal or stable human impacts as defined by the presence of agricultural and urban areas, roads and a high population density, and the presence of significant flow structures (Whitfield et al., 2012). RHBN-N was created to provide a nationally balanced network suitable for national studies. Similarly to the HCDN-2009 dataset, a minimum data availability of 20 years of almost continuous streamflow records was required for a station to qualify".

In line with the previous comment: line 85 references a screening approach by Whitfield et al 2012 for Canada but it would be interesting for the reader to know if the classification of catchment with or without regulated catchments is comparable to the one of the USGS data set

See our response above.

Figure 1 d): to clarify, this shows all stations of SCDNA, even the ones that were not used in the study? As only precipitation data is considered for the manuscript, would it not be clearer to only show the incorporated stations?

Upon verification, this was actually not the case and all SCDNA stations contain precipitation data. This sentence was removed from the revised manuscript. We still generated new maps for Fig. 1 in order to distinguish SWE, precipitation, and streamflow stations, and river basins actually used for the analysis presented in this study, and edited the figure caption accordingly.

Figure 1c): some of the SWE data products seem to overlap in some locations. Was the data for these locations compared to get a general feeling of the SWE data and its quality? Just curious.

In several provinces and territories of Canada, such as in Alberta, British Columbia and the Yukon, co-located automated snow stations (snow pillows and scales) and manual snow surveys are collected. While the pillows/scales are in more open areas, the surveys tend to be a mix of open and forested. The co-located measurements usually agree well, but are not identical. It should however be noted that the automatic and manual data have a different spatial representativeness and report at a different temporal frequency – i.e., the snow surveys consist of multi-point manual data collected along a given transect.

Part of the issue could also be the scale of the map that cannot reproduce the variability of the snow measurements network in terms of position and elevation. For example, the Alberta Government mountain snow surveys are 20-50 km apart, so they would appear as overlapping circles on the map. There are some surveys that are in close proximity, but at different elevations, so they capture SWE changes across an elevation gradient. There are also academic research sites/snow surveys that are in close proximity to the Alberta surveys. This might be resolved to a certain extent with the new maps created for Fig. 1, including only stations actually used in this study (see our response above).

In terms of data quality, snow monitoring experts and data providers follow standard operating procedures, which include quality assurance (QA)/quality control (QC) protocols. In addition to the quality standards applied by all the different data providers, a systematic QC procedure is described in Vionnet et al. (2021b; cited in the manuscript) and has been applied to all the snow data used in this study, with the exception of the Pacific Northwest National Laboratory SNOTEL bias-corrected and quality-controlled (BCQC) dataset.

We have added some text to the manuscript for readers who may be similarly curious (L121-128): "All SWE data used for this study were quality controlled (QC). In addition to the quality standards applied by the different data providers, a systematic QC procedure

is described in Vionnet et al. (2021b) and was applied to all the snow data used in this study, with the exception of the already bias-corrected and quality-controlled SNOTEL dataset. Several SWE stations appear to be overlapping on Fig. 1c. In various Canadian provinces and territories like Alberta, British Columbia, and the Yukon, automated snow stations and manual snow surveys are collected at the same sites. While the measurements from these stations generally agree, they are not identical due to micro-scale spatial variability. In addition, the stations overlap may partly be due to the scale of the map which does not allow to accurately display the variability of the snow measurements network in terms of position and elevation".

Figure 2: this is a very informative and well designed overview figure! I would suggest referencing it more often in the manuscript (e.g. the volume aggregation of the target periods in line 169)

Thank you, we're glad you like the figure! This is a great idea and we added references to the figure throughout the methods sub-sections, when we thought the figure would be particularly useful to understand the text.

Section 2.1.1*, line 127: while it is mentioned that the SWE and streamflow data is used regardless of whether the years have complete records (due to the following gap filling process, max allowable gap length listed as 15days in line 164) I wonder whether there was a limit of how much missing data was seen as acceptable in total? A few days per year or even a few weeks or months throughout the total record? Figure A2 in the Appendix suggests that some were heavily gap filled compared to the original timeseries?

*Note that there is a typo and the reviewer refers to Section 2.2.1 here, and not Section 2.1.1. The same is true for the reviewer comment below.

No total maximum allowable gap length was applied to sub-select stations. For the streamflow data, we expected the datasets to be nearly complete due to the quality checks done in the production of the HCDN-2009 and RHBN datasets, where 20 years of (near) complete streamflow data were required for a station to qualify (see answer above). For the SWE data, several stations had large data gaps and may indeed have been heavily gap filled during the quantile mapping gap filling step. Selecting a maximum gap length threshold would require additional thorough analysis of the gap length vs. the quality of the gap filling, and this was not done as the gap filling is not the focus of this manuscript.

However, we did explore the impact of the length of the window used for gap filling the SWE data (from +/- 1 to +/- 7 days) on the quality of the artificial gap filling. The findings indicate that extending the window to +/- 7 days yielded greater benefits by significantly increasing the amount of filled gaps in the data, while having a fairly low impact on the quality of the filled data. From this, we infer from this that setting a maximum threshold

for allowable missing data would excessively reduce the number of SWE stations (since PCA requires a complete dataset) and adversely affect the quality of the hindcasts.

We now discuss the impact of this methodological decision in the text (in the SWE pre-processing Section 2.2.3; L238-241) and added this point as a topic that warrants further attention in the "Workflow developers" discussion (Section 4.4; L602-605): "It is important to note that no threshold was set to define a total maximum allowable gap length for each station. Consequently, certain stations may have undergone substantial gap filling, as can be seen on Fig. A2. However, we speculate that setting such a threshold would have been counterproductive, as it would have significantly decreased the number of SWE stations available as predictors, thereby affecting the quality of the hindcasts produced", and "Subsequent studies could investigate how various methodological choices influence the quality and the effectiveness of the gap filling, using the artificial gap filling function. This could involve examining the consequences of implementing a total maximum allowable gap length to sub-select stations, or adjusting the window used for the gap filling through quantile mapping".

Section 2.1.1, line 145 and section 2.2.2: gap filling through linear interpolation, could the authors elaborate on potential limitations of this approach for both streamflow and SWE? And the potential consequences of those limitations for the regime classification approach using the streamflow as well as for defining the statistics for the CDF construction in case of the SWE gap filling using quantile mapping later?

The linear interpolation could potentially have consequences for both the streamflow and the SWE gap-filled data. Regarding streamflow, it might have resulted in missed flow peaks, especially for smaller basins with higher response times. However, employing three metrics for peak flow events identification and nival basin selection, and the selection of the circular statistics method with a regularity threshold of 0.65, could have potentially mitigated some of these effects. Furthermore, the linear interpolation step was deemed essential for the streamflow dataset to ensure effective filling of the data gaps. As for SWE, it could have affected the construction of CDFs for donor and target stations during the quantile mapping step, possibly resulting in inaccuracies in the gap-filled data. However, we speculate that using a station's own data for gap filling via temporal interpolation might yield better results that using another station's data, especially given the relatively gradual temporal variations in SWE.

We have added some text to discuss this in the revised manuscript (L183-187 and L241-245): "The streamflow linear interpolation could have impacted the regime classification, leading to missed flow peaks, especially for smaller river basins with faster response times. Nevertheless, all stations had nearly complete datasets, as this was a requirement for selection in the creation of both datasets (see Section 2.1.1). Furthermore, the use of three metrics for peak flow event identification, coupled with the utilization of the circular statistics method with a regularity threshold of 0.65, potentially mitigate some of these issues", and "linear interpolation might have impacted the construction of CDFs for donor

and target stations, possibly introducing inaccuracies into the gap-filled data. Nevertheless, we speculate that utilizing a station's own data for gap filling via temporal interpolation could yield superior results compared to utilizing data from other stations, especially given the relatively gradual temporal variations in SWE".

Section 2.2.3, line 190: for clarification, the 'original' SWE data gets gap filled twice in different ways? First by linear interpolation to be able to get the statistics for the CDF construction and then the 'original' SWE data gets gap filled with a separate quantile mapping approach again? Or were there specific values that were not be able to be gap filled before?

The SWE data was first gap filled with linear interpolation prior to constructing the CDFs. The gap filled data were then gap filled through quantile mapping, as the linear interpolation could not fill all the gaps in this dataset.

We have clarified this in the revised manuscript, referring readers to the graphical methods (L212-213): "After applying linear interpolation, we then utilize quantile mapping to fill the remaining gaps using data from neighbouring stations (see Fig. 2)".

Was there a specific reason (other than that SWE is used for the PCA) that streamflow did not undergo the same two step gap filling process as SWE (linear and then quantile mapping)?

Streamflow did not have as many gaps, again due to the quality checks done in the production of the HCDN-2009 and RHBN datasets, and a one-step process was deemed sufficient for this variable.

This sentence was added to the revised manuscript (L195-197): "Due to the data availability quality checks conducted during the production of the HCDN-2009 and RHBN streamflow datasets, a one-step gap filling process was considered sufficient for streamflow, in contrast to the two-step gap filling performed for SWE (see Section 2.2.3)".

Section 2.2.4, line 212: for clarification, "comprising ten years for training the regression model and an additional year for generating the hindcast, using the leave-one-out cross-validation approach."

this is the leave-one-out cross-validation approach definition?

This is indeed the definition of the leave-one-out cross-validation, with additional information on the minimum number of years required for training the model.

We rephrased the sentence and added a definition of the leave-one-out cross-validation to clarify the text. The revised text is (L253-256): "We use a leave-one-out cross-validation approach for forecasting, whereby each data point in the dataset is sequentially withheld

as a validation set, while the model is trained on the remaining data points. We require a minimum of eleven years of overlapping data in total, comprising ten years for training the regression model and an additional year for generating the hindcast".

Section 2.2.4, line 215: are the total 11 years used for the PC or the split dataset (10-1)? Line 226 refers to the first but just to check

The PCA is done for the total 11 years, after which the dataset is split for forecasting. We discuss the effect of these methodological decisions compared to operational forecasting approaches on L469-477 of the original manuscript.

Section 2.2.4, line 224: "We conduct a PCA and fit a new model for each predictor-predictand combination" – does this mean an OSL model for every target period? Or just one OSL model for per location for all target periods?

Hindcasts are indeed generated using an independent regression model for each river basin, initialization date, target period, and year left out. This is mentioned in Section 2.2.4, on L224-225 of the original manuscript: "We conduct a PCA and fit a new model for each predictor-predictand combination". This was also noted by Reviewer 1.

We have now clarified this point in the text (L280-281): "An independent regression model is used to produce an ensemble hindcast for each river basin, initialization date, target period, and year left out".

Section 2.2.5, line 240: for clarification: target periods listen in line 169 are not the same as the 'periods of interest' introduced in this line? And the verification will be on the 'periods of interest' or the initial introduced target periods? (KGE result description suggests the latter)

Parts of the results are shown for all target periods, namely the KGE" decomposition, the reliability index, and the CRPSS boxplots. The CRPSS maps and the ROC AUC (Fig. 7 and 8) show results for the 'periods of interest' only.

We clarified the text in Section 3.3 (L372-373) to: "The Fair CRPSS maps show results for each basin's period of interest only, in order to be able to compare results across river basins for a single lead time". The text in Section 3.4 (L389-391) was changed to: "Unlike plots for the KGE" and its decomposition, the reliability index, and the Fair CRPSS (boxplots only), these plots show results for each basin's period of interest only".

Section 2.2.5, line 241: with every nival basin potentially having different 'periods of interest' does this have an effect on the hindcast verification if general or averaged results over the 62 stations are presented as not every 'period of interest' has the same number of samples?

This is a very good point and the varying sample sizes may indeed affect the variability or "noise" in the results.

We have added a new subplot that shows the number of basins in each boxplot (i.e., for each period of interest – target period combination). We refer to this new subplot in the text so this information is transparent. The text now reads (L409-412): "As these plots show results for each basin's period of interest only, some of the boxplots have limited data points (i.e., the number of data points in each boxplot is shown in Fig. 8c). This could explain some of the differences between boxplot span and the variability or noise observed in each subplot".

Table 1 once again a very informative and clear table that I am sure many readers will appreciate!

Thank you!

Figure 4 and corresponding description (line 280-285): as Figure 4 is the first figure in that specific presentation style it might be nice for the reader to get a more in depth guide how to interpret it for the different target periods and the lead periods presented (despite lines 280-285, there was still some confusion when analyzing it the first time)

We agree that this is a useful addition, thank you for the suggestion. The improved text now reads (L324-329): "Figure 4 shows the hindcast performance in terms of the Kling-Gupta Efficiency (KGE") and its decomposition into correlation, variability, and bias in the different subplots. In each subplot, results are shown for each hindcast target period (coloured lines), as a function of hindcast initialization dates (x-axis). Looking at the KGE" for hindcasts produced for the target period September $1^{st}$ to $30^{th}$ (purple line) as an example, we observe the evolution in performance over time, from hindcasts initialized on $1^{st}$ January (left-most dot) to those initialized on $1^{st}$ September (right-most dot). The hindcasts' lead time decreases progressively from left to right within each subplot".

section 4: nice to a see a refreshing take on a discussion

Thank you, we're glad you enjoyed it.

section 4.2: maybe a reference back to both hypotheses in line 244 would be good to remind the reader of them

Good idea, we now refer readers back to Section 2.2.5 when mentioning the two hypotheses in discussion section 4.1 and 4.2.

the presented work is focusing on catchments with limited regulations and the discussion includes a separate focus on decision-makers: do the authors think that this work can also

be helpful for decision-makers (e.g. water managers) working in more regulated catchments?

This question is also based on the explanation in section 2.2.2 in line 160, where it is stated that streamflow was "converted into volumes that capture the spring freshet and that may be of interest of water users (e.g., for water supply management, hydropower generation, irrigation scheduling, early warnings of floods and droughts)".

Or is there a category of catchments that would fall in between non regulated and regulated where the suggested probabilistic framework could still work?

This is an interesting suggestion. This study focused solely on forecasting the streamflow of unregulated rivers. Regulation in the sense here, alters the relationship between the hydro-meteorological drivers of streamflow and streamflow. In those regulated river basins, it would still be valuable to predict streamflows upstream of the regulation (e.g., the inflows to a reservoir, streamflows upstream of a city, or of a regulated river), where predictability comes from upstream SWE stations, for water management decision-making downstream (e.g., for hydropower generation, flood early warning, riverine transportation). These forecasts can also be useful in the context of naturalized flows, whereby the streamflow without regulation or abstraction is of interest and needed for decision-making.

We have added some reflection on this topic to the revised manuscript, in the "Decision-makers" discussion (Section 4.3; L524-531): "This study focused solely on forecasting streamflows in unregulated river basins, which may include river basins upstream of a regulation, such as a reservoir or an urbanized area. Regulation alters the relationship between the hydro-meteorological drivers of streamflow and streamflow. In those regulated river basins, it is however still valuable to predict streamflows upstream of the regulation (e.g., the inflows to a reservoir, streamflows upstream of a city, or of a regulated river segment), where predictability comes from upstream SWE stations, for water management decision-making downstream (e.g., for water supply management, hydropower generation, irrigation scheduling, early warnings of floods and droughts, riverine transportation). This methodology could additionally add value in regulated catchments where the naturalized flow is used for water management decision-making".

Other changes made

We also made additional modifications to the manuscript, GitHub and Zenodo repositories, which are listed below. Beyond making minor clarifying edits and typo corrections throughout the manuscript, we made the following specific changes:

- We made a minor adjustment to Fig. 2 (in the upper box, step 1) to change "overlapping SWE-Q data" to "overlapping SWE and Q data", as this was confusing.

- We made minor adjustments to the Fig. 3 subplots. For Fig. 3a, the marker size was increased slightly to match the marker style of the new Fig. 1 maps. For Fig. 3b, the y label was capitalized to match the style of other manuscript figures.

- We added mentions of Table 1, giving an overview of the metrics used to assess the quality of the hindcasts, throughout the results so that readers can easily refer back to the equations used.

- We changed the inset histograms in Fig. 7 to ensure a regular bin width and we kept the y-axis range the same across all histograms (now showing y-axis values) to make the figure clearer.

- We rephrased a paragraph in the "Forecasters" discussion (Section 4.2; L498-501) to avoid overstating the findings of this paper and to encourage further analysis aimed at identifying spatial patterns in the hindcast skill and their relationships with the physical processes of runoff generation. Including these results as well would make the paper too lengthy. The revised text reads: "Figure 7 illustrates higher and longer predictability in interior and western North American river basins, contrasting with lower and shorter predictability in the north and in the east., which partly aligns with the findings of Zheng et al. (2018). However, further analysis is needed to identify spatial patterns in the hindcast skill and their relationships with the physical processes of runoff generation".

- We added a reference to the now publicly available SWE dataset from Colleen and Vionnet (2024), which compiles snow survey data in the USA used in this study: Mortimer, C. and Vionnet, V.: Northern Hemisphere historical in-situ Snow Water Equivalent dataset (1979-2021), https://doi.org/10.5281/ZENODO.10287092, 2024

- We adjusted the plot dimensions of the Fig. 9 panels.

- We are in the process of making minor changes to the GitHub and Zenodo repositories to facilitate their use. These should be available shortly and the new versions will be added to the manuscript prior to its publication.